# ERα promotes murine hematopoietic regeneration through the Ire1α-mediated unfolded protein response

Richard H Chapple[1], Tianyuan Hu[1], Yu-Jung Tseng[2], Lu Liu[3], Ayumi Kitano[1], Victor Luu[1], Kevin A Hoegenauer[1], Takao Iwawaki[4], Qing Li[3], Daisuke Nakada[1,2]*

[1]Department of Molecular & Human Genetics, Baylor College of Medicine, Houston, United States; [2]Graduate Program in Translational Biology and Molecular Medicine, Baylor College of Medicine, Houston, United States; [3]Division of Hematology/Oncology, Department of Medicine, University of Michigan, Ann Arbor, United States; [4]Division of Cell Medicine, Department of Life Science, Medical Research Institute, Kanazawa Medical University, Ishikawa, Japan

**Abstract** Activation of the unfolded protein response (UPR) sustains protein homeostasis (proteostasis) and plays a fundamental role in tissue maintenance and longevity of organisms. Long-range control of UPR activation has been demonstrated in invertebrates, but such mechanisms in mammals remain elusive. Here, we show that the female sex hormone estrogen regulates the UPR in hematopoietic stem cells (HSCs). Estrogen treatment increases the capacity of HSCs to regenerate the hematopoietic system upon transplantation and accelerates regeneration after irradiation. We found that estrogen signals through estrogen receptor α (ERα) expressed in hematopoietic cells to activate the protective Ire1α-Xbp1 branch of the UPR. Further, ERα-mediated activation of the Ire1α-Xbp1 pathway confers HSCs with resistance against proteotoxic stress and promotes regeneration. Our findings reveal a systemic mechanism through which HSC function is augmented for hematopoietic regeneration.

DOI: https://doi.org/10.7554/eLife.31159.001

*For correspondence:
nakada@bcm.edu

Competing interests: The authors declare that no competing interests exist.

## Introduction

In order to adapt to changing animal physiology or tissue damage, tissue stem cells must respond to long-range signals emitted from other organs (*Nakada et al., 2011*). For instance, stem cells in the intestine, central nervous system, hematopoietic system, and germline are regulated by insulin and nutritional status (*Chell and Brand, 2010*; *Cheng et al., 2014*; *LaFever and Drummond-Barbosa, 2005*; *Lehtinen et al., 2011*; *McLeod et al., 2010*; *O'Brien et al., 2011*; *Sousa-Nunes et al., 2011*; *Yilmaz et al., 2012*), and the circadian rhythm affects skin and hematopoietic stem cells (*Janich et al., 2011*; *Méndez-Ferrer et al., 2008*). Age-related changes in blood-borne factors regulate muscle and neural stem cell aging (*Brack et al., 2007*; *Conboy et al., 2005*; *Katsimpardi et al., 2014*; *Sinha et al., 2014*; *Villeda et al., 2011*). However, few long-range signals, such as prostaglandin E2 (*North et al., 2007*) or those that are modulated by prolonged fasting (*Cheng et al., 2014*), have been found to augment the ability of tissue stem cells to repair damaged tissues. Understanding how long-range signals regulate tissue stem cell function may pave the way to harness stem cells for regenerative medicine.

The hematopoietic system has a remarkable regenerative potential demonstrated by the ability of a single hematopoietic stem cell (HSC) to rebuild the entire hematopoietic system upon transplantation, and to regenerate the tissue upon damage, such as those caused by chemo- or radio-therapies. A network of factors operating cell intrinsically or extrinsically promote regeneration by HSCs

(*Mendelson and Frenette, 2014*). Cell-extrinsic factors produced by the HSC niche, such as the stem cell factor (*Zsebo et al., 1992*), angiogenin (*Goncalves et al., 2016*), pleiotrophin (*Himburg et al., 2010*), EGF (*Doan et al., 2013*), or Dkk1 (*Himburg et al., 2017*) have particular therapeutic potential as they may be used as radioprotectors or radiomitigators to accelerate the recovery of the hematopoietic system (*Citrin et al., 2010*). Despite the extensive interest in augmenting the regenerative potential of HSCs, much remains to be investigated regarding the factors that promote hematopoietic regeneration by HSCs, and the mode of action of these factors.

The female sex hormone estrogen is an emerging soluble factor involved in specification and self-renewal of HSCs. Estradiol (E2), the most potent endogenous estrogen, is predominantly produced by the granulosa cells of the ovaries and regulates a diverse array of tissues, such as the reproductive organs, bone, breast, brain, and the hematopoietic system (*Edwards, 2005*). During zebrafish development, maternal E2 stored in the yolk promotes the specification of HSCs (*Carroll et al., 2014*). In mice, E2 produced by the ovaries stimulates division of HSCs via the estrogen receptor α (ERα). Exogenous E2, when administered at a physiological level similar to the level observed during pregnancy, also promotes division of HSCs in male mice, indicating that the E2-ERα pathway activation is sufficient induce HSC division (*Nakada et al., 2014*). A selective estrogen receptor modulator tamoxifen also promoted HSC division similar to E2 (*Sánchez-Aguilera et al., 2014*). Interestingly, even though E2 significantly increases HSC divisions, unlike infection or interferon responses it does not lead to HSC depletion (*Baldridge et al., 2010*; *Esplin et al., 2011*; *Essers et al., 2009*; *Passegué et al., 2005*; *Sato et al., 2009*). This raises a question of whether HSC function is impaired by E2 at the cost of increased division, or whether E2 elicits a protective mechanism to maintain HSCs while stimulating HSC division.

The unfolded protein response (UPR) is an evolutionarily conserved mechanism that promotes protein homeostasis (proteostasis) upon various stresses by increasing the expression of protein foldases and chaperones (*Hetz et al., 2013*; *Janssens et al., 2014*; *Wang and Kaufman, 2012*). Three pathways — the PERK-eIF2α-CHOP pathway, the ATF6 pathway, and the IRE1-XBP1 pathway — coordinately regulate the UPR in mammals. Gpr78 (also called BiP), a chaperone localized in the endoplasmic reticulum (ER), tethers the three ER stress sensors (PERK, ATF6, and IRE1) in inactive states during homeostasis. Upon ER stress, Gpr78 disassociates with PERK, ATF6, and IRE1 to allow them to become activated and induce the UPR. The UPR plays essential roles in highly secretary cells, such as plasma cells and pancreatic β-cells that have high protein flux through the ER. In addition, recent evidence established that the UPR is also involved in a wide range of pathological states, since many stress signals impinge upon the ER (*Malhotra and Kaufman, 2007*). Interestingly, evidence in *C.elegans* indicates that the UPR can be extrinsically activated, potentially mediated by an as yet unidentified neurotransmitter (*Sun et al., 2012*; *Taylor et al., 2014*; *Taylor and Dillin, 2013*). Whether the UPR in mammalian tissue stem cells is regulated by systemic factors remains elusive.

Here we demonstrate that the female sex hormone E2 increases the regenerative capacity of HSCs upon transplantation, and improves bone marrow and peripheral blood recovery after irradiation. ERα, in response to E2 stimulation, activated a protective UPR by inducing the expression of Ire1α in HSCs. The Ire1α-Xbp1 branch of the UPR augmented proteotoxic stress resistance in HSCs and promoted regeneration. Our results reveal that the UPR in HSCs can be modulated by systemic factors, extending the systemic activation of the UPR to tissue stem cell biology.

## Results

### Estradiol promotes the regenerative capacity of HSCs

To address the question of whether E2 stimulation affects HSC function, we first performed colony-forming assays by sorting single HSCs (*Figure 1—figure supplement 1A*, for gating strategy) from oil- or E2-treated male mice into methylcellulose media. We used male mice unless otherwise noted since estrogen levels fluctuate in females during the estrus cycle. HSCs from E2-treated animals exhibited a greater proportion of immature colonies containing granulocytes, erythrocytes, macrophages, and megakaryocytes (gemM) compared to HSCs from oil-treated mice (*Figure 1A*), suggesting that E2 increases the multipotency of HSCs. To quantify the effects of E2 in promoting megakaryocytic potential of HSCs we used a collagen-based media that enables outgrowth and enumeration of megakaryocytes. Although freshly isolated HSCs were incapable of forming any colonies

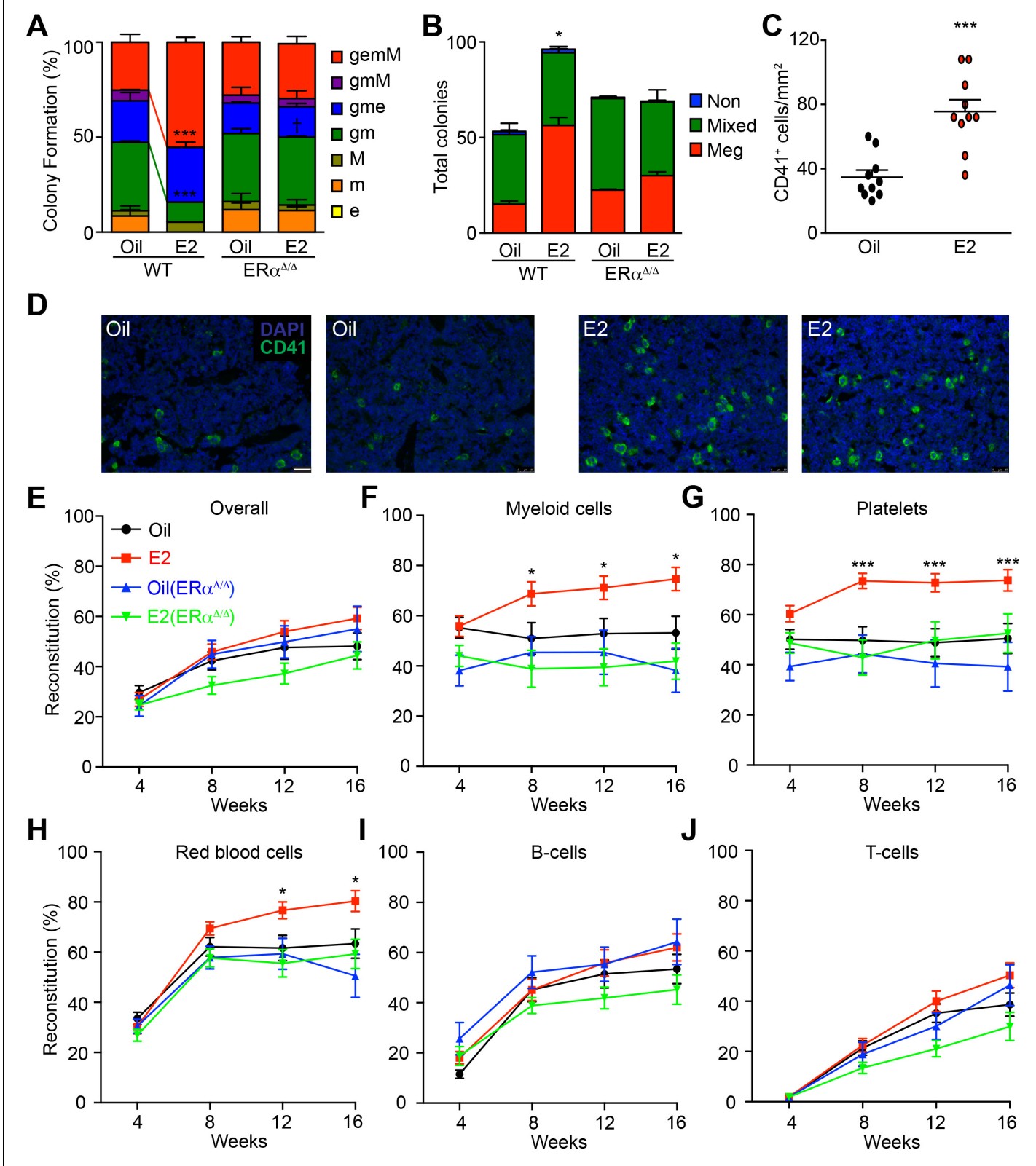

**Figure 1.** Estrogen Enhances the Myeloid Potential of HSCs. (**A**) Colony formation by single HSCs from control or E2 treated *Esr1*<sup>fl/fl</sup> (WT) and *Mx1-Cre; Esr1*<sup>fl/fl</sup> (Esr1<sup>Δ/Δ</sup>) mice (96 wells per animal, n = 4 assays/group). Colonies were collected, cytospun, and scored after Wright Giemsa staining. gemM: granulocyte, erythroid, monocyte, megakaryocyte; gmM: granulocyte monocyte megakaryocyte; gme: granulocyte, monocyte, erythroid; gm: granulocyte, monocyte; M: megakaryocyte; m: monocyte; e: erythroid. Red and green lines indicate significant interaction of treatment and genotype of

*Figure 1 continued on next page*

Figure 1 continued

gemM and gm, respectively (***p<0.001, ANOVA) (B) Megakaryocyte differentiation potential in collagen-based MegaCult assays (n = 6, three independent experiments, two technical replicates per experiment). Meg, colonies containing exclusively megakaryocytes as indicated by cholinesterase staining; Mixed, colonies containing both megakaryocytes and other myeloid cells; Non, colony with no megakaryocytes. *p<0.05, ANOVA. (C) Numbers of CD41$^+$ megakaryocytes as indicated by immunofluorescent staining of bone marrow sections (n = 10, 5 fields of view per section). (D) Representative images of CD41 stained bone marrow sections from oil- and E2-treated mice. Scale bar represents 50 µm. (E–J) Levels of donor (GFP$^+$) engraftment in recipient mice that were transplanted with 100 GFP$^+$ HSCs (oil- or E2-treated *Ubc-GFP; Esr1$^{fl/fl}$* (n = 4 donors each, 24 and 26 recipients respectively) or *Ubc-GFP; Mx1-Cre; Esr1$^{fl/fl}$* (n = 3 donors each, 14 and 23 recipients respectively) and 2 × 10$^5$ competitor GFP$^-$ cells. Each panel represents donor chimerisms in (E) overall mononuclear cells, (F) Mac-1/Gr-1$^+$ myeloid cells, (G) CD41$^+$ FSC/SSC$^{low}$ platelets, (H) Ter119$^+$ FSC/SSC$^{low}$ red blood cells, (I) B220$^+$ B-cells, and (J) CD3$^+$ T-cells. All data represent mean ±standard deviation; *p<0.05; **p<0.01; and ***p<0.001 by Student's t-test unless otherwise noted.

DOI: https://doi.org/10.7554/eLife.31159.002

The following figure supplement is available for figure 1:

**Figure supplement 1.** (A) Gating strategy for the identification of HSCs and MPPs in both oil- and E2-treated mice.
DOI: https://doi.org/10.7554/eLife.31159.003

in this media, potentially due to the lack of HSC supportive cytokines, HSCs after a brief culture in media containing cytokines exhibited robust megakaryocytic differentiation in this system. We found that HSCs isolated from E2-treated mice not only formed more colonies but they also exhibited a significantly increased capacity to form colonies containing megakaryocytes (*Figure 1B*). Consistent with the increased megakaryopoiesis by E2, we observed significantly more megakaryocytes in the bone marrow of E2-treated mice than oil-treated mice (*Figure 1C–D*). These results indicate that E2 treatment enhances the clonogenic potential of HSCs towards myeloid, erythroid, and megakaryocytic lineages.

We then tested whether E2 treatment affects multilineage reconstitution of HSCs. To this end, we competitively transplanted 100 purified HSCs isolated from male *Ubc-GFP* mice that were treated with either oil or E2 for one week. Compared to oil-treated controls, HSCs isolated from E2-treated mice exhibited enhanced reconstitution of Mac-1/Gr-1$^+$ myeloid cells, Ter119$^+$ red blood cells, and CD41$^+$ platelets (*Figure 1E–H* and *Figure 1—figure supplement 1B–C*). Interestingly, HSCs from E2-treated mice did not exhibit reduced lymphoid cell reconstitution (*Figure 1I–J*), indicating that the enhanced myeloid/erythroid/megakaryocytic cell reconstitution by E2 stimulated HSCs was not at the cost of lymphoid cell reconstitution. Next, we ascertained if E2 affects long-term self-renewal capacity by performing secondary transplantation. Reconstituted bone marrow derived from E2-treated HSCs gave rise to stable engraftment across all lineages in secondary recipients (*Figure 1—figure supplement 1D*). Therefore, although E2 treatment increases HSC division (*Nakada et al., 2014*), it does not negatively affect long-term self-renewal capacity of HSCs but rather increases HSC function.

To determine whether ERα expressed in hematopoietic cells is responsible for the effects of E2 on HSC function, we treated *Mx1-Cre; Esr1$^{fl/fl}$* mice (and control littermates) with poly I:C to delete ERα from the hematopoietic system (hereafter described as ERα-deficient mice). ERα-deficient mice and control littermates were then treated with E2 for one week followed by colony assays to examine HSC function. The increased ability of E2-stimulated HSCs to form immature colonies (*Figure 1A*) and megakaryocytes (*Figure 1B*) was abolished by deleting ERα, indicating that ERα promotes HSC function upon E2 stimulation.

Further, we determined whether the increased capacity of HSCs to reconstitute the hematopoietic system upon transplantation was dependent on ERα. 100 HSCs isolated from ERα-deficient and control littermates treated with either oil or E2 were transplanted along with competitor cells. ERα-deficient HSCs without E2 stimulation exhibited intact long-term multilineage reconstitution (*Figure 1E–J*), indicating that ERα is dispensable for steady state HSC function, as shown previously (*Thurmond et al., 2000*). However, ERα-deficient HSCs did not exhibit increased myeloid/erythroid/megakaryocytic cell reconstitution upon E2 treatment as observed in wild-type HSCs (*Figure 1F–H*). These results establish that the E2-ERα pathway enhances functions within the hematopoietic cells to promote myeloid/erythroid/megakaryocytic reconstitution by HSCs in response to E2.

## Estradiol promotes hematopoietic recovery after irradiation

The results presented above demonstrate that E2 promotes hematopoietic regeneration by HSCs in a transplantation model. As another model of hematopoietic regeneration, we examined the effects of E2 on hematopoiesis after injury imposed by total body irradiation (TBI) (*Doan et al., 2013*; *Goncalves et al., 2016*; *Himburg et al., 2017*; *Himburg et al., 2010*). We treated male mice with oil or E2 for one week, and irradiated them with a sublethal dose of 600 cGy (*Figure 2—figure supplement 1* for experimental scheme). Serial sampling of peripheral blood after irradiation revealed that E2 treatment prior to irradiation significantly accelerated the rate at which platelets and white blood cells recover, whereas the effects on red blood cell counts were modest (*Figure 2A–C*). To determine whether ERα is responsible for the effects of E2 on hematopoietic regeneration after irradiation, we administered poly I:C-treated *Mx1-Cre; Esr1fl/fl* mice with E2 followed by sublethal irradiation. The accelerated recovery of RBC, PLT, and WBC in E2-treated mice was not observed in ERα-deficient mice, indicating that ERα is required for E2 to promote regeneration (*Figure 2D–F*). These results indicate that E2 treatment promotes hematopoietic recovery after stress, consistent with our prior finding that E2 promotes the reconstitution potential of HSCs.

The positive effects of E2 on hematopoietic recovery improved the survival of mice after irradiation. Male mice were treated with oil or E2 for one week, and irradiated with a semi-lethal dose of 800 cGy. Whereas 89% of oil-treated mice died within 28 days after irradiation, E2 treatment significantly reduced the fraction (to 29%) of mice succumbing to irradiation-induced lethality (*Figure 2G*). This increased survival is consistent with an early study that described that female mice are more radioresistant than male mice, which we confirmed here (*Figure 2H*), and that the estrus cycle affects the radiosensitivity of female mice (*Rugh and Clugston, 1955*). These results suggest that the positive effect of E2 on hematopoietic regeneration improves the survival of mice after hematopoietic injury.

## Estradiol accelerates hematopoietic progenitor cell regeneration after ionizing radiation

Our observation that E2 treatment accelerated the recovery of peripheral blood cells prompted us to examine whether E2 improves hematopoietic stem/progenitor cell (HSPC) function after irradiation. E2 treatment significantly accelerated the recovery of whole bone marrow (WBM) cells at days 5, 7, and 14 after irradiation (*Figure 3A*), consistent with the accelerated recovery of blood cells. We then examined whether E2 affects HSPC frequency after irradiation. Irradiation caused rapid and prolonged depletion of immature lineage⁻Sca-1⁺c-kit⁺ (LSK) cells for up to two weeks post irradiation ($0.17 \pm 0.04\%$ before irradiation versus $0.02 \pm 0.02\%$ at two weeks after irradiation). Intriguingly, E2 treatment prior to irradiation significantly increased the frequency of LSK cells compared to control oil-treated mice 14 days after TBI (*Figure 3B* and *Figure 3—figure supplement 1A*). Combined with the increased numbers of WBM cells in E2-treated irradiated mice, E2 treatment induced a 6.3-fold expansion (p<0.01) of the size of LSK pool at day 14 (*Figure 3C*), suggesting that E2 treatment prior to TBI promotes LSK regeneration. E2 treatments every other day post-TBI also increased hematopoietic cell numbers after irradiation (*Figure 3—figure supplement 1B–E*). In contrast, E2 treatment did not significantly affect the frequency of lineage⁻Sca-1⁻c-kit⁺ committed myeloid progenitors (MP), but did increase the total numbers of these cells (*Figure 3—figure supplement 1F–H*). We assessed apoptosis rates in lineage⁻ BM cells shortly after irradiation exposure by Annexin V staining and observed significantly lower numbers of Annexin V⁺ cells in E2 treated cohorts at both days 3 and 5 post-TBI (*Figure 3—figure supplement 1I*). These results suggest that the preservation of HSPC numbers by E2 treatment could be a function of enhanced proliferation (*Nakada et al., 2014*), protection from irradiation-induced apoptosis, or a combination of both. Collectively, these results suggest that E2 not only enhances recuperation of peripheral blood cells after irradiation (*Figure 2A–C*), but also enhances hematopoietic regeneration by promoting the recovery of immature HSPCs.

Next, we examined whether E2 affects HSCs regeneration after irradiation. Since irradiation compromised the expression of several HSC markers (data not shown and (*Simonnet et al., 2009*), we accessed long-term reconstitution capacity as a surrogate of HSC function by transplanting WBM cells from irradiated animals that had been treated with control oil or E2 into wild-type recipients. Cells from irradiated, oil-treated mice exhibited negligible reconstitution upon transplantation

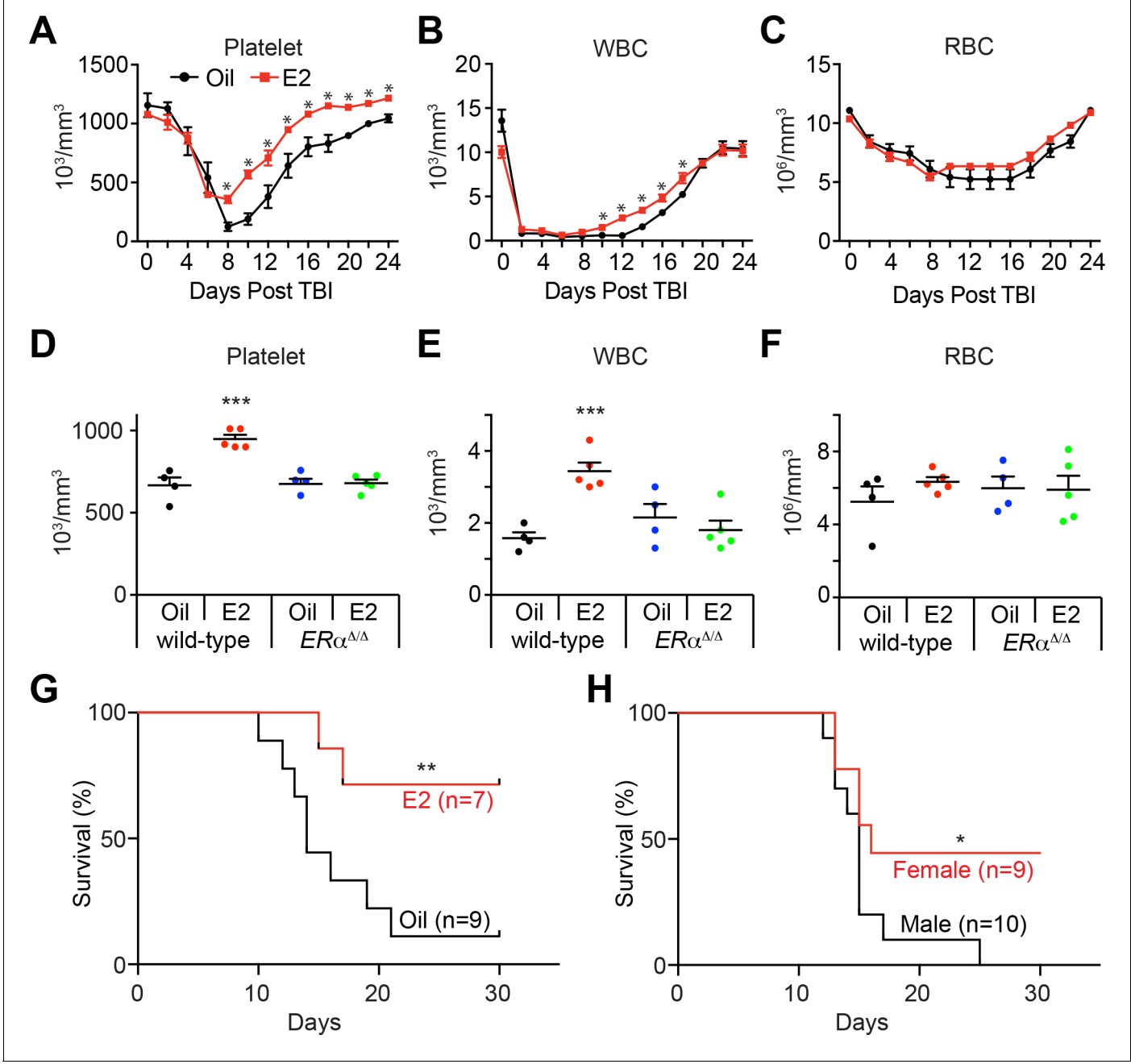

**Figure 2.** Estrogen Promotes Hematopoietic Recovery After Irradiation. (A–C) Mean PB counts of (A) platelets, (B) white blood cells, and (C) red blood cells of oil- and E2-treated mice (black and red lines, respectively. n = 4 Oil, n = 5 E2, two independent experiments). *p<0.05 by Student's t-test. (D–F) Mean PB counts of (D) platelets, (E) white blood cells, and (F) red blood cells of control or *ERα*-deficient mice, treated with either oil or E2. Data were collected 14 days after TBI. ***p<0.001, ANOVA. (G) Survival curves of C57BL/6 male mice treated with oil or E2 for 1 week followed by 800cGy TBI. **p<0.01 by a log-rank test. (H) Survival curves of C57BL/6 male and female mice following 800cGy TBI. *p<0.05 by a log-rank test. All data represent mean ±standard deviation.

DOI: https://doi.org/10.7554/eLife.31159.004

The following figure supplement is available for figure 2:

**Figure supplement 1.** Experimental design of hematopoietic regeneration following TBI.
DOI: https://doi.org/10.7554/eLife.31159.005

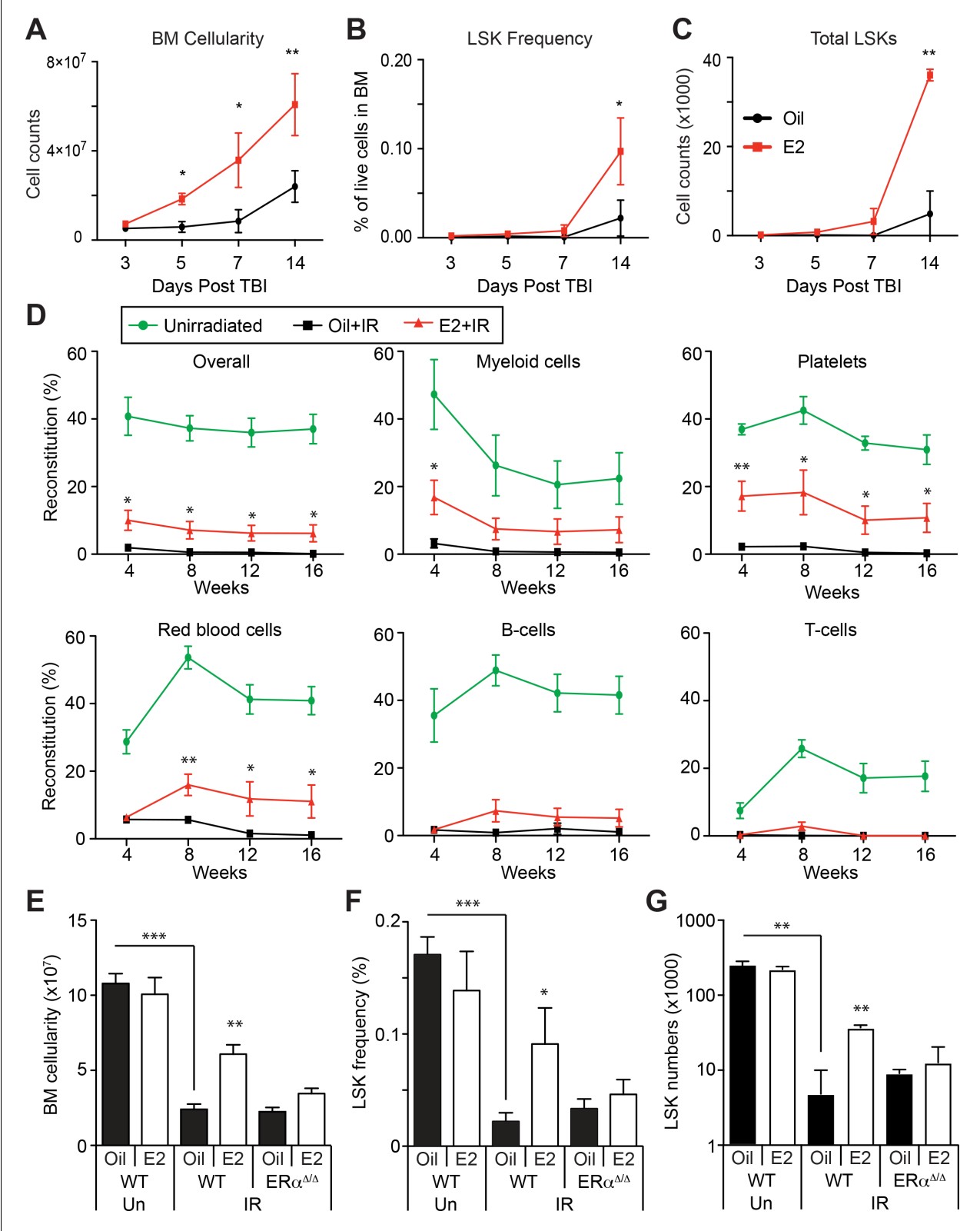

**Figure 3.** Estrogen Promotes HSPC Regeneration After Irradiation. (**A**) Bone marrow cellularity (**A**), LSK frequencies (**B**), and total numbers of LSK cells (**C**) were analyzed at 3, 5, 7, and 14 days after 600cGy of TBI. E2 treatment significantly accelerated the recovery of bone marrow cellularity, and the frequencies and the total numbers of LSK cells. (**B–C**) Frequencies (**B**) and total numbers (**C**) of LSK cells in the bone marrow drop significantly 14 days after 600cGy of irradiation; this affect is rescued by E2 treatment in wild-type mice but not in *ERα*-deficient mice (n = 7 over four independent

*Figure 3 continued on next page*

*Figure 3 continued*

experiments). (D) Competitive transplantation assays using $5 \times 10^5$ WBM cells from Ubc-GFP$^+$ unirradiated mice (green lines) or sublethally irradiated, oil-treated mice (black lines) or sublethally irradiated E2-treated mice (red lines) along with competitor cells (n = 8, two donors, four recipients per donor). E2 pretreatment significantly rescued the ability of WBM cells to reconstitute the myeloid cells, platelets, and red blood cells after irradiation. (E–G) Bone marrow cellularity (E), LSK frequencies (F), and total numbers of LSK cells (G) in the bone marrow are significantly lower 14 days after 600cGy of irradiation (IR) compared to unirradiated mice (Un). This effect is rescued by E2 treatment in wild-type mice but not in *ERα*-deficient mice (n = 7, four independent experiments). All data represent mean ±standard deviation; *p<0.05; **p<0.01; and ***p<0.001 by Student's t-test in (A–D), and by ANOVA in (E–G).

DOI: https://doi.org/10.7554/eLife.31159.006

The following figure supplement is available for figure 3:

**Figure supplement 1.** Regeneration of HSPCs after irradiation by E2 stimulation.

DOI: https://doi.org/10.7554/eLife.31159.007

(*Figure 3D*). Strikingly, we observed that cells from mice that were treated with E2 prior to irradiation demonstrated a preservation of long-term reconstitution potential in the myeloid, erythroid, and platelet lineages — a feature that reflects HSC activity (*Oguro et al., 2013*; *Pietras et al., 2015*; *Sanjuan-Pla et al., 2013*; *Yamamoto et al., 2013*) — although their lymphoid-reconstitution potential was impaired. These results demonstrate that E2 treatment improves the regeneration of the hematopoietic system, augmenting the function of cells with long-term myeloid reconstitution capacity.

We then tested whether ERα regulates HSPC regeneration after irradiation. Control and ERα-deficient mice were treated with oil or E2, sublethally irradiated, and analyzed by flow cytometry. We found that deleting ERα abolished the accelerated recovery of bone marrow cellularity after irradiation in response to prior E2 treatment (*Figure 3E*). Furthermore, whereas E2 treatment promoted the regeneration of LSK cells in the bone marrow after irradiation, it failed to do so in ERα-deficient mice (*Figure 3F,G*). These results indicate that ERα, which is highly expressed in HSPCs, promotes hematopoietic regeneration in response to E2 stimulation.

## Estradiol activates the unfolded protein response (UPR) pathway in HSCs

To understand the mechanism through which E2 increases HSC function, we employed RNA-seq to profile the gene expression changes of HSCs upon E2 treatment. Control and *ERα*-deficient male mice were treated with oil or E2 for one week, and HSCs were isolated for RNA-seq. Unsupervised hierarchical clustering analysis revealed that wild-type HSCs from E2-treated mice clustered separately from other groups, indicating that E2 induces a transcriptional response that is dependent on ERα (*Figure 4A*). We then performed gene set enrichment analysis (GSEA) to identify gene signatures enriched in HSCs following E2 treatment. The GSEA revealed that Myc and E2F target genes and genes involved in cell cycle regulation were enriched in E2-treated HSCs (*Figure 4B* and *Figure 4—figure supplement 1A–H*), consistent with our finding that E2-treated HSCs are more proliferative (*Nakada et al., 2014*). Several Myc target genes were induced by E2 treatment in vitro (*Figure 4—figure supplement 1B–H*), although c-Myc, N-Myc, and L-Myc were not induced in HSCs by E2 as determined by RNA-seq (data not shown). Additionally, several megakaryocyte lineage genes were increased in E2 treated HSCs (*Figure 4—figure supplement 1I*). Interestingly, we found that a gene set representing the unfolded protein response (UPR) was also upregulated in HSCs in response to E2 (*Figure 4B Figure 4—figure supplement 1J*). We focused specifically on the UPR gene set and examined whether the enrichment of the UPR gene set upon E2 treatment depends upon ERα. To this end, we performed GSEA in a pairwise fashion comparing the four HSC datasets with each other. We found that wild-type HSCs from E2-treated mice have significant (p<0.005) enrichment of UPR related genes compared to all other treatment groups, including E2-treated ERα-deficient HSCs (*Figure 4C*). This analysis indicates that E2 treatment induces the UPR in HSCs in an ERα-dependent manner. Importantly, we found that the expression of *Ern1*, which encodes a key regulator of the UPR, Ire1α, was significantly increased in E2-treated HSCs (*Figure 4A and D*). The increased expression of *Ern1* was also evident in our previously published microarray analysis (GDS4944) (*Nakada et al., 2014*). Moreover, multiple chaperones were induced in HSCs by E2

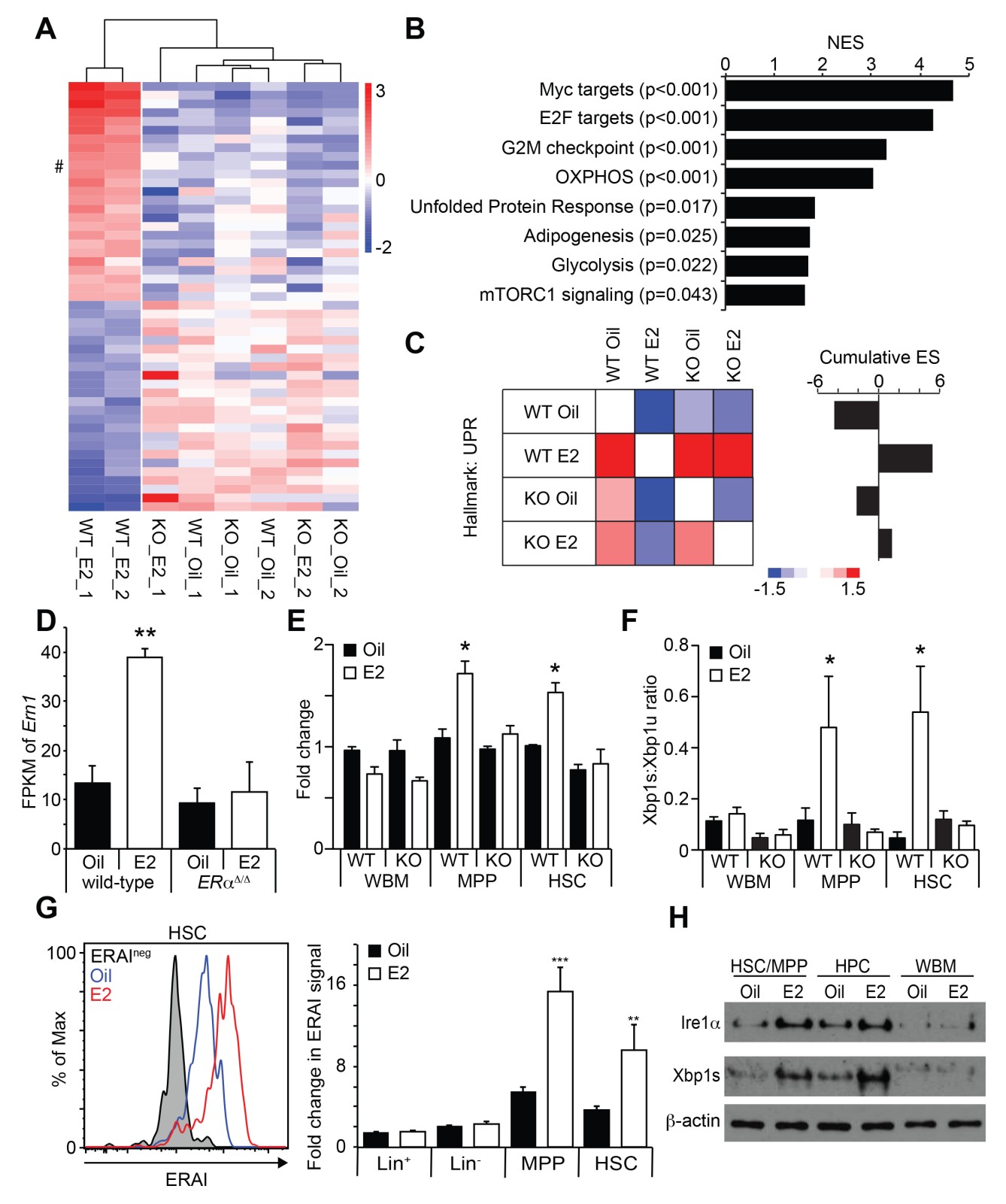

**Figure 4.** Estrogen Activates the Ire1α-Xbp1 Branch of the UPR. (**A**) Unsupervised hierarchical clustering of genes with log2 fold change larger than one identified by RNA-seq. Top 25 genes for each cluster are shown. # indicates *Ern1*, which encodes Ire1α. (**B**) GSEA was performed between oil- and E2-treated HSCs. Gene sets significantly enriched in E2-treated HSCs are shown. (**C**) Pair wise GSEA was performed to test the enrichment of UPR-related genes in wild-type or *ERα*-deficient HSCs after either oil- or E2-treatment (left). E2-treated wild-type HSCs had enrichment of UPR genes, as indicated

*Figure 4 continued on next page*

*Figure 4 continued*

by the cumulative enrichment score (right). (**D**) Fragments per kilobase of exon per million fragments mapped (FPKM) of *Ern1* transcript was increased in HSCs upon E2 treatment, and this increase was dependent on ERα. \*\*p<0.01 by two-way ANOVA. (**E**) Quantitative PCR assays confirmed that the induction of *Ire1α* in HSCs, as well as in MPPs, upon E2 treatment was dependent on ERα (n = 3, two independent experiments). \*p<0.05 by two-way ANOVA. (**F**) *Xbp1* splicing was determined by a two-color Taqman assay. *Xbp1* splicing was induced in HSCs and MPPs upon E2 treatment in an ERα-dependent manner. The ratio between spliced (Xbp1s) to unspliced (Xbp1u) Xbp1 transcript is shown (n = 4, two independent experiments). \*p<0.05 by two-way ANOVA. (**G**) *Xbp1* splicing was also determined by flow cytometry using the ERAI strain. Grey histogram represents the background signal from ERAI⁻ HSCs. \*\*p<0.01; and \*\*\*p<0.001 by Student's t-test. (**H**) Immunoblotting was performed using HSC/MPP (CD48⁻ᐟˡᵒʷLSK), progenitor cells (CD48⁺LSK), and WBM cells. Both Ire1α and Xbp1s protein levels were increased by E2 treatment in immature cells. All data represent mean ±standard deviation.

DOI: https://doi.org/10.7554/eLife.31159.008

The following figure supplements are available for figure 4:

**Figure supplement 1.** E2 treatment increases the expression of megakaryocyte-related genes and chaperones in HSCs.

DOI: https://doi.org/10.7554/eLife.31159.009

**Figure supplement 2.** Ire1α induction in HSCs is not a function of increased HSC proliferation.

DOI: https://doi.org/10.7554/eLife.31159.010

(*Figure 4—figure supplement 1K*). These results suggested that the UPR was increased in HSCs after E2 exposure.

Three branches, governed by the PERK-eIF2α-CHOP, ATF6, and IRE1-XBP1 pathways, regulate the UPR. Each of these pathways controls the expression of multiple UPR target genes, consisting of chaperones and genes involved in metabolism (*Hetz et al., 2013*). To better understand how E2 activates UPR in HSCs, we performed qRT-PCR on key mediators of the UPR pathway. Consistent with our RNA-seq analysis, we confirmed that the induction of *Ern1* in HSCs upon E2 treatment was fully dependent on ERα (*Figure 4E*). However, the components of the two other UPR branches, such as *Ddit3* (encoding CHOP), *Eifak3* (encoding PERK), and, *Atf6*, were unaffected by E2 treatment (*Figure 4—figure supplement 2C*), suggesting that the Ire1α pathway is activated upon E2 stimulation. To test whether Ire1α was activated in a lineage-biased subset of HSCs we examined Ire1α expression in CD48⁻ᐟˡᵒʷ LSK cells fractionated based on CD150 expression levels (*Figure 4—figure supplement 2A,B*) (*Beerman et al., 2010*). Ire1α levels were increased consistently across each subset indicating that E2 acts on the entirety of the HSC pool.

Ire1α is a bifunctional protein with a serine-threonine kinase domain and an RNase domain. In response to ER stress, the RNase activity of Ire1α becomes activated and splices *Xbp1* mRNA (*Cox and Walter, 1996*; *Yoshida et al., 2001*). This *Xbp1* splicing event produces a functional Xbp1s protein that serves as a transcriptional regulator critical for the induction of UPR response genes (*Hetz et al., 2013*; *Janssens et al., 2014*). We thus tested whether *Xbp1* splicing is induced upon E2 treatment in HSCs using a quantitative PCR method that detects both the unspliced and spliced forms of *Xbp1*. We found that *Xbp1* splicing was induced in response to E2 treatment in both HSCs and MPPs, but not in WBM cells, suggesting that immature hematopoietic progenitor cells are more sensitive to E2 treatment to activate Xbp1 (*Figure 4F*). This was confirmed using the *ERAI* strain, in which *Xbp1* splicing can be monitored by Venus fluorescence (*Iwawaki et al., 2004*). Flow cytometry analysis of cells isolated from *ERAI* mice treated with oil or E2 revealed that *Xbp1* splicing, as determined by Venus fluorescence, was significantly increased by E2 treatment in HSCs and MPPs, but not in lineage⁻ and WBM cells (*Figure 4G*). Importantly, *Xbp1* splicing was entirely dependent upon *ERα*, consistent with our finding that the E2-ERα pathway induces *Ire1α* expression (*Figure 4F*). We also confirmed that the protein levels of Ire1α and Xbp1s were increased upon E2 treatment in both CD48⁻ᐟˡᵒʷ (containing HSCs and MPPs) and CD48⁺ fractions of the LSK population, but not in WBM cells (*Figure 4H*). Taken together, these results indicate that the Ire1α-Xbp1 branch of the UPR is activated in E2-treated HSCs. As increased protein synthesis is associated with proliferation and may activate the UPR, we tested whether E2-induced activation of the Ire1α-Xbp1 pathway is a general feature of activated HSCs. Using a protocol adapted from previous reports (*Liu et al., 2012*; *Schmidt et al., 2009*; *Signer et al., 2014*), we tested whether E2 treatment affected protein translation rates in HSCs. Consistent with a previous report (*Signer et al., 2014*), HSCs and MPPs had significantly lower protein translation compared to whole bone marrow cells (*Figure 4—figure supplement 2D*). However, the protein synthesis rate of HSCs did not significantly change upon E2

treatment (*Figure 4—figure supplement 2D*), consistent with a previous report demonstrating that cycling HSCs do not have increased protein translation (*Signer et al., 2016*). Moreover, inducing HSCs to proliferate (either by G-CSF or poly I:C administration) did not increase *Ire1α* expression (*Figure 4—figure supplement 2E,F*), and these treatment had limited effects on the expression of chaperone genes (*Figure 4—figure supplement 2G*) compared to the broad effects E2 had on chaperones (*Figure 4—figure supplement 1K*). Thus, activation of the Ire1α-Xbp1 pathway is not a general response used by HSCs upon cell cycle activation.

## Ire1α is a direct target of ERα in HSPCs

Genome wide ERα binding profiles have been extensively studied in other cell types, but ERα binding sites in hematopoietic cells remain elusive. Moreover, profiling the genome wide ERα binding sites in rare HSCs is technically challenging. To begin to understand the changes in transcriptional factor landscapes induced by E2 treatment in purified HSCs, we profiled the chromatin accessibility sites in oil- or E2-treated HSCs using ATAC-seq (*Buenrostro et al., 2013*). We reasoned that if ERα is indeed a major regulator of E2-induced transcriptional changes, we would observe changes in chromatin accessibility in response to E2 with ERα footprints, which could be discovered by the presence of estrogen response elements (ERE). 5,000 HSCs were isolated from oil- or E2-treated mice, and the isolated nuclei were subject to an ATAC-seq protocol. We observed highly concordant ATAC-seq profiles across samples, identifying 51,230 overlapping peaks between all replicates (*Figure 5—figure supplement 1*). We identified a subset of peaks that were differentially upregulated in E2-treated cells (*Figure 5A*). Both the de novo and the supervised motif search identified ERE as the most enriched motif found in differentially regulated peaks, consistent with the idea that ERα is the major regulator of the transcriptional changes induced by E2 in HSCs (*Figure 5B*). Of interest, one of the differential peaks present in E2-treated HSCs was located approximately 2.5 kb upstream of the TSS of *Ire1α* (termed peak 2) (*Figure 5C*). Further sequence interrogation of this locus revealed that this ATAC-seq peak contains a ERE motif with one nucleotide mismatch (GGTCAnnnTGACa) from the consensus ERE (*Figure 5C*, lower panel). Given that ERE motifs found in prototypical ERα target genes, such as Oxytocin and pS2, contain one base mismatch from the consensus (*Gruber et al., 2004*), we next functionally validated the ERE in this locus. To this end, we first tested whether the DNA sequence of this region have E2- and ERα-responsive transcriptional activity. We cloned the 4 kb region of *Ern1* into a luciferase reporter plasmid, transfected into 293 T cells with or without a plasmid expressing ERα, and treated the cells with E2 or control vehicle and measured luciferase activity. This revealed that the *Ern1*(−4 kb) construct has E2- and ERα-responsive transactivating capacity (*Figure 5D*). Importantly, this E2- and ERα-responsive transactivating capacity was diminished when the ERE was mutated in the *Ern1*(−4 kb) construct, indicating that the ERE located upstream of *Ern1* is functional. To determine if ERα binds to this locus in primary hematopoietic progenitor cells, we performed ChIP-qPCR. c-kit[+] HSPCs were isolated from oil- or E2-treated mice and were subject to ChIP. We found that the ERα occupancy at peak 2 increased significantly upon E2 treatment (*Figure 5E*). This result coincides with our finding that this locus becomes more accessible upon E2 treatment and that it contains a functional ERE. These results establish that *Ire1α* is a direct target of ligand-activated ERα in hematopoietic progenitor cells.

## Estradiol promotes protective unfolded protein response in HSCs

The UPR was recently shown to regulate cellular responses in HSPCs upon proteotoxic stresses (*Miharada et al., 2014*; *van Galen et al., 2014*). Upon ER stress, human HSCs upregulated the proapoptotic PERK branch of the UPR, thus causing cell death, whereas downstream progenitors upregulated the pro-survival IRE1α-XBP1 branch to survive (*van Galen et al., 2014*). We hypothesized that E2 stimulation upregulates the pro-survival Ire1α-Xbp1 branch in HSCs, thus endowing them with resistance to various cellular stressors. To test this theory, we induced ER stress in vitro in freshly isolated LSK cells from oil- or E2-treated mice with the drugs tunicamycin and thapsigargin. Cells were then analyzed for viability using Annexin V staining and colony assays. Both tunicamycin and thapsigargin increased the frequencies of Annexin V[+] LSK cells (*Figure 6—figure supplement 1A,B*) and reduced clonogenicity (*Figure 6—figure supplement 1C*). However, in vivo E2 treatment significantly rescued these effects, indicating that HSPCs exposed to E2 are protected from

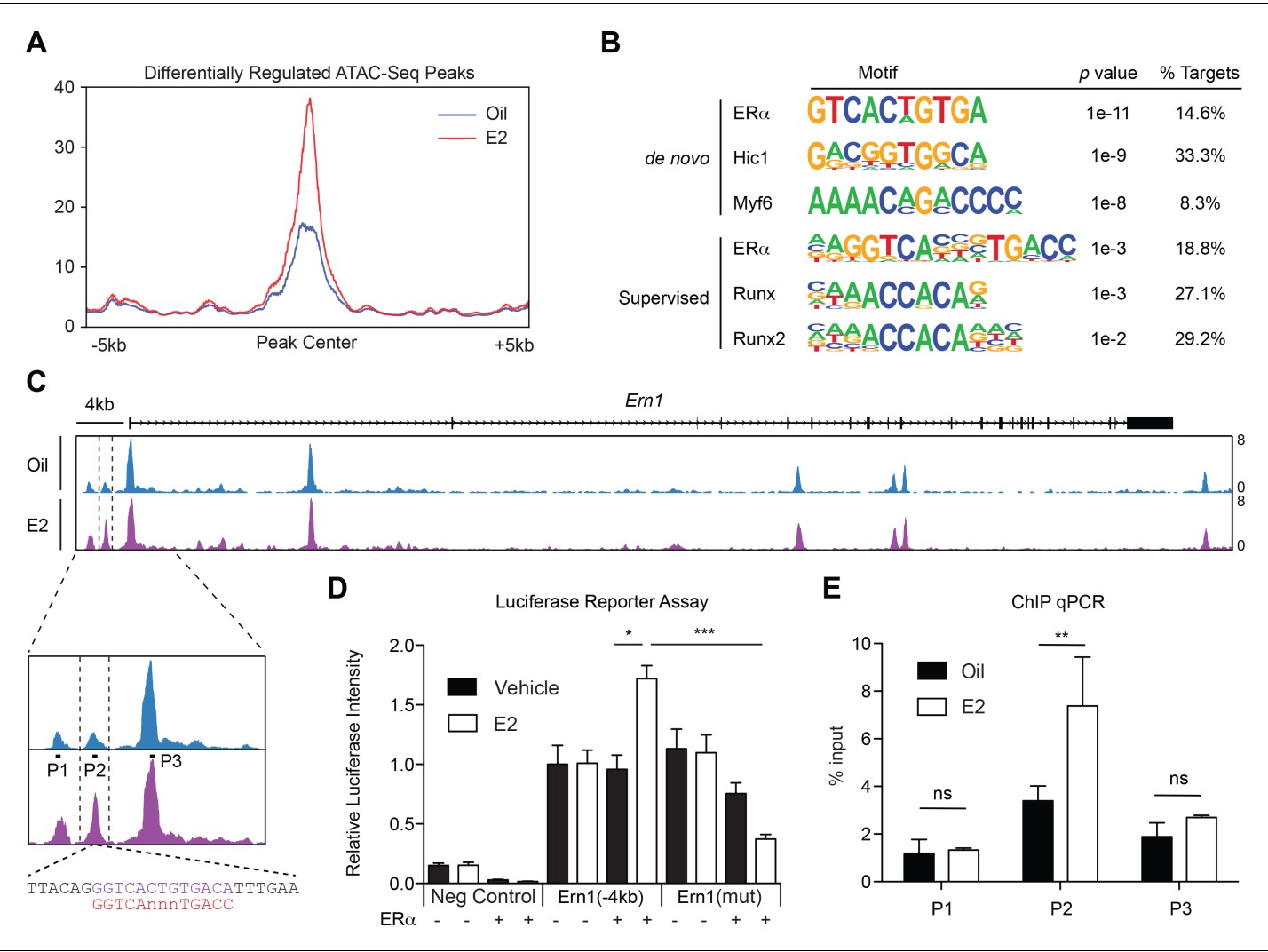

**Figure 5.** ERα Associates with the *Ire1α* Upstream Locus in HSPCs (**A**) Metagene analysis indentifying differentially expressed peaks in E2 treated HSPCs. (**B**) Homer motif analysis. Both de novo (unsupervised) and supervised motif-identifying algorithms classified EREs as the most enriched motif in E2-treated HSPCs. (**C**) Representative gene track of *Ire1α* with peaks identified from ATAC-Seq. Three peaks were identified upstream of the *Ire1α* promoter (denoted Peak 1, 2, and 3). Peak two was identified as differentially upregulated in E2 treated cells, and contains a ERE motif (lower panel, purple) with one nucleotide mismatch with the consensus ERE (shown in red). Primers were designed for each peak and are denoted P1, P2, and P3. (**D**) Luciferase reporter assay. A 4 kb genomic sequence of *Ire1α* regulatory elements containing peaks 1–3 was cloned into pGL2 Basic plasmid. 293 T cells were transfected with the reporter plasmids, ERα expressing or an empty vector, and a control renilla luciferase plasmid. Cells were treated with E2 for 20 hr and luciferase activity measured. A significant increase in the luciferase activity was observed with the Ire1α construct in an E2- and ERα-dependent manner, and this response was abolished by mutating the ERE. (n = 4 independent experiments, two technical replicates per condition) ***p<0.001; *p<0.05, ANOVA (**E**) ChIP qPCR was conducted for the three peaks identified by ATAC-Seq using primers shown in (**C**). y-axis represents the relative value to percent input in Oil or E2 treated cells. (n = 4; two independent experiments, two biological replicate) **p<0.01; Student's t-test.
DOI: https://doi.org/10.7554/eLife.31159.011

The following figure supplement is available for figure 5:

**Figure supplement 1.** Heatmaps of ATAC-seq peaks discovered in two biological replicates of HSC samples isolated from oil- and E2-treated mice.
DOI: https://doi.org/10.7554/eLife.31159.012

proteotoxic stress. These results establish that the E2-ERα pathway confers resistance against ER stress, paralleling the activation of the protective Ire1α-Xbp1 branch of the UPR.

Increased oxidative stress, such as that caused by ionizing irradiation, can cause protein oxidation, leading to accumulation of protein aggregates, which in turn elicits ER stress and activates the UPR (*Berlett and Stadtman, 1997*; *Malhotra and Kaufman, 2007*; *Squier, 2001*). We thus

examined whether the total body irradiation we used in the previous experiments affected proteostasis of hematopoietic cells. We quantified protein aggregates using a recently published protocol (*Sigurdsson et al., 2016*), and observed a dose-dependent increase in the levels of protein aggregates following sublethal doses of irradiation (*Figure 6A*). Interestingly, E2-treatment significantly reduced the levels of protein aggregates after TBI in LSK cells but not WBM cells, in concordance with the induction of Ire1α and splicing of *Xbp1* (*Figure 6B*).

We then tested whether the Ire1α-Xbp1 branch is required for the E2-ERα pathway to endow HSCs with stress resistance. Using our recently described mouse and human HSPC genome editing protocol (*Gundry et al., 2016*), we designed multiple single-guide RNAs (sgRNAs) against Ire1α, and electroporated them as a ribonucleoprotein (RNP) complex with Cas9 protein into LSK cells. By immunoblotting and sequencing, we identified sgRNAs that efficiently ablated Ire1α (*Figure 6C* and *Figure 6—figure supplement 1D*). These sgRNAs were used in subsequent experiments. We isolated LSK cells from both oil- and E2-treated mice, and electroporated them with Cas9-RNPs with sgRNAs against *Ire1α* or control *Rosa26*. Clonal analysis of electroporated LSK cells indicated that about 25% of the clones had deletions between the two sgRNA target sites, while another 50% of the clones had small indels detected by TIDE analysis (*Figure 6—figure supplement 1D*). The bulk-modified cells were then plated in media with or without ER stressors. Strikingly, we found that deleting *Ire1α* abolished the resistance endowed by E2 against ER stressors (*Figure 6D*). These results demonstrate that E2 promotes ER stress resistance by activating the cytoprotective UPR branch governed by the Ire1α-Xbp1 pathway.

Finally, we tested whether the induction of the Ire1α-Xbp1 pathway by ERα promotes hematopoietic regeneration. LSK cells were electroporated with Cas9-RNP targeting either Rosa26 or Ire1α and transplanted into lethally irradiated mice. Recipients were treated with control oil or E2 every other day, and peripheral blood was collected at day 14 post-TBI. Consistent with our previous findings (*Figure 2D–F*), recipients of Rosa26-edited LSK cells demonstrated enhanced hematopoietic recovery when treated with E2, which was largely suppressed when Ire1α was deleted (*Figure 6E*). We also investigated the regeneration of HSPCs in the bone marrow of these recipient mice. The frequencies of LSK and myeloid progenitor cells were significantly increased in mice receiving E2 treatment after transplantation (*Figure 6F–G* and *Figure 6—figure supplement 1E–F*). However, this effect was abolished in similarly treated recipients of Ire1α-edited LSK cells. Collectively, these results demonstrate that the E2-ERα module activates and at least partly depends on the Ire1α-Xbp1 pathway to promote hematopoietic regeneration.

## Discussion

Our results establish that E2 activates the Ire1α-Xbp1 branch of the UPR in HSCs through ERα, thus augmenting hematopoietic regeneration. E2 treatment increased the capacity of HSCs to reconstitute the hematopoietic system upon transplantation and accelerated hematopoietic regeneration after total body irradiation. E2-ERα signaling transcriptionally induced the expression of Ire1α, which then promoted the splicing of *Xbp1* mRNA to induce Xbp1s, a transcription factor critical for the UPR. The importance of the Ire1α-Xbp1 pathway-mediated UPR is exemplified by our finding that E2 augments hematopoietic recovery after transplantation through Ire1α. Thus, E2 is a soluble long-range signal that promotes hematopoietic regeneration at least partly by inducing a cytoprotective UPR in HSCs.

Whether activation of the UPR is beneficial or detrimental for HSC function appears to depend on which pathway becomes activated and to what extent it is activated. Uncontrolled activation of the UPR, particularly of the PERK pathway, has been shown to impair HSC function. Deletion of Grp78 (BiP) in mice activates all three branches of the UPR and reduces the HSC pool, causing lymphopenia (*Wey et al., 2012*). Human HSCs activate the PERK pathway upon proteotoxic stress, rendering them susceptible to apoptosis (*van Galen et al., 2014*). On the other hand, activation of the Ire1α-Xbp1 branch or the protein folding capacity improves HSC function. While human HSCs activate the PERK pathway upon proteotoxic stress and undergo apoptosis, downstream progenitors activate the Ire1α-Xbp1 pathway to survive (*van Galen et al., 2014*). Additionally, overexpression of a molecular chaperone ERDJ4 (encoded by *DNAJB9*) protects human HSCs from proteotoxic stress and increases the reconstitution capacity (*van Galen et al., 2014*). Our results are consistent with the model that activation of the Ire1α-Xbp1 branch by E2 protects HSCs from proteotoxic stress and

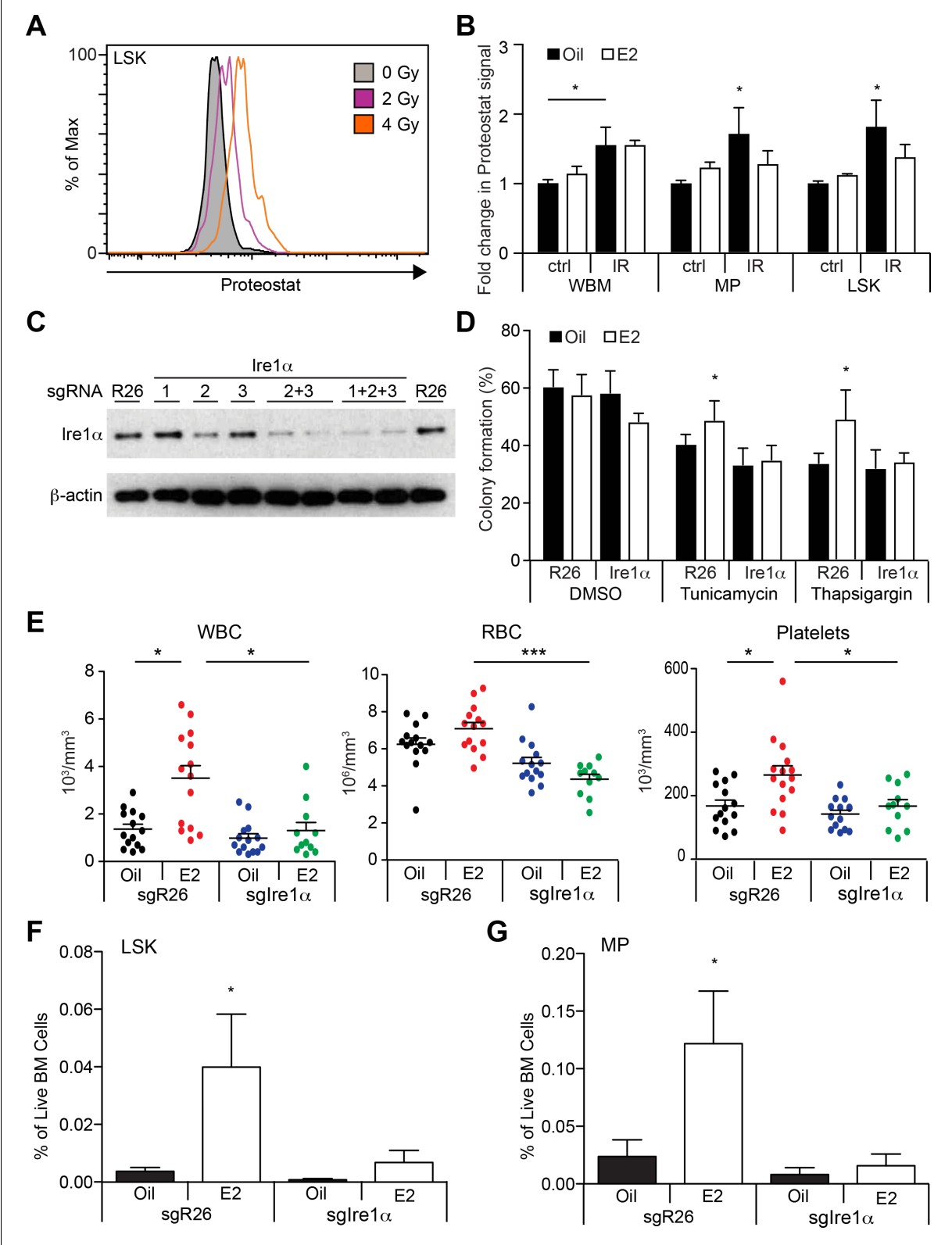

**Figure 6.** Estrogen Mitigates Proteotoxic Stress in HSPCs (**A**) Representative FACS plots of Proteostat staining in LSK cells from mice subjected to various doses of radiation. (**B**) Proteostat staining revealed that irradiation induces protein aggregation in all hematopoietic cell types, and E2 reduces aggregation in immature progenitor cells. Mice were treated with oil or E2 for 1 week followed by a single dose of radiation (IR: 400cGy) or unirradiated (ctrl), and analyzed 16 hr later by flow cytometry (n = 5, three independent experiments). *p<0.05, two-way ANOVA (**C**) c-kit+ progenitor cells were

*Figure 6 continued on next page*

*Figure 6 continued*

electroporated with Cas9 protein with sgRNAs against either *Rosa26* (R26) or three different sgRNAs against *Ire1α*. Cells were collected after 48 hr and analyzed by immunoblotting. (D) Deletion of *Ire1α* by the CRISPR/Cas9 system sensitized LSK cells to ER stressors. LSK cells from oil- or E2-treated mice were gene edited as in (C), incubated overnight with ER stressors, and plated on semi-solid media for colony formation. (n = 3 independent experiments) *p<0.05, two-way ANOVA (E) 2000 gene-edited LSK cells (*Rosa26* (n = 14, Oil; n = 14, (E2) or *Ire1α* (n = 14, Oil; n = 11, E2; four independent experiments) were transplanted into lethally irradiated recipients, followed by oil or E2 administration every other day. Shown are blood counts 14 days after transplantation. *p<0.05; ***p<0.001; Kruskal Willis with Dunn's multiple comparisons test. (F) LSK and (G) myeloid progenitor (MP) frequencies were measured in recipients of gene-edited LSKs. (n = 5, two independent experiments) *p<0.05, two-way ANOVA. In (E–G), Ire1α sgRNA 2 + 3 or 1 + 2 + 3 was used with similar results. All data represent mean ±standard deviation.
DOI: https://doi.org/10.7554/eLife.31159.013

The following figure supplement is available for figure 6:

**Figure supplement 1.** Ire1α is Necessary for Hematopoietic Regeneration in E2 Treated Mice.
DOI: https://doi.org/10.7554/eLife.31159.014

promotes hematopoietic regeneration, although we cannot formally exclude the possibility that two other UPR branches (PERK and ATF6) are also involved. IRE1 also promotes decay of mRNA localized to the ER, called regulated IRE1-dependent decay of mRNA (RIDD) (*Maurel et al., 2014*). However, RIDD seems unlikely to be activated in E2-treated HSCs since translation is not attenuated in HSCs after E2 exposure. Furthermore, RIDD is usually associated with apoptotic cell death, which is not observed in E2-treated cells. It is important to note that Xbp1 is largely dispensable for HSC maintenance during homeostasis (*Bettigole et al., 2015*), indicating that although Xbp1 is not required for the homeostatic maintenance of HSCs, its activation provides additional protection against proteotoxic stress. Thus, selective activation of the Ire1α-Xbp1 pathway is key to harnessing the UPR to improve hematopoietic regeneration. How E2 preferentially activates the Ire1α-Xbp1 branch without appreciable activation of the PERK and ATF6 pathways remains to be explored.

Consistent with our previous finding that E2 promotes division of HSCs (*Nakada et al., 2014*), we found that E2-treated HSCs had increased expression of genes known as Myc target genes, some of which were induced in vitro after E2 stimulation. It is important to note, though, that although some of the Myc targets were induced, c-Myc, N-Myc, or L-Myc were not induced by E2. Thus, to what extent the global upregulation of Myc-target genes in HSCs after E2 treatment reflects Myc-induced responses or an indirect consequence of the activated cell cycle remains unclear. More work is needed to determine the direct target of the E2-ERα module that promotes HSC division.

Our findings indicate that E2 promotes hematopoietic regeneration by protecting HSPCs from irradiation-induced stress. An earlier study reported that female mice are more resistant to acute irradiation than male mice, and that the estrus cycle affects radioresistance (*Rugh and Clugston, 1955*), although careful interpretation is required since strain differences also contribute to radioresistance (*Grahn, 1958*). Estrogenic compounds such as genistein have also been known to have radioprotective effects (*Ha et al., 2013*). Additionally, it has been demonstrated that engraftment of human HSPCs is more efficient in female recipient mice than male mice (*Notta et al., 2010*). The mechanism behind this sexual difference in HSC engraftment remains to be established, but it is tempting to speculate that the female hormonal milieu protects engrafting HSCs from stress. Further investigation on how estrogens, or more broadly hormones, prepare the recipient to accommodate transplanted HSCs, including the involvement of the bone marrow niche, may pave the way to improve the efficacy of bone marrow transplantation. Since estrogens also have detrimental hematopoietic side-effects, such as causing thromboembolism (*Tchaikovski and Rosing, 2010*), future studies should aim to separate the beneficial effects of estrogens on HSC function from other negative effects.

In concordance with the effect of E2 on HSPC regeneration, E2 also improved the blood cell counts after irradiation or transplantation, although the effects of E2 on mature cells in the blood were less pronounced than the effects on HSPCs. We postulate that since production of mature cells involves multiple intermediate cell stages during differentiation and that it is also affected by a variety of homeostatic mechanisms (such as thrombopoietin production by the liver and the kidney), the end output in the blood may not be as robust as the effects on HSPCs.

More broadly, our data extend the role of systemic UPR activation to regeneration by adult stem cells. Exciting new data have described that the UPR is a key pathway in intercellular and inter-tissue

communication regulating tissue homeostasis and aging. Chemical chaperones, such as osmolytes and hydrophobic compounds, aid in cellular protein folding capacity to maintain proteostasis, and have therapeutic potential in treating metabolic and neurodegenerative diseases (*Cortez and Sim, 2014*). Interestingly, taurocholic acid, a chemical chaperone that is more abundant in the bile acid of fetal liver compared to adult liver, inhibits protein aggregation in fetal liver HSCs to support their rapid expansion during development and in culture (*Miharada et al., 2014*; *Sigurdsson et al., 2016*). This illustrates that systemic factor(s) can affect the protein folding capacity of HSCs, although it remains unclear whether bile acid affects the UPR of HSCs in adult animals. In nematodes and fruit flies, impaired proteostasis of a cell type or a tissue can communicate to other cells to coordinate stress responses systemically by upregulating the UPR in the periphery (*Berendzen et al., 2016*; *van Oosten-Hawle and Morimoto, 2014*). Additionally, activation of the UPR in neurons by Xbp1s expression extends lifespan of *C.elegans*, and strikingly, augments the UPR in the intestine through a cell non-autonomous mechanism involving an unknown neurotransmitter (*Taylor and Dillin, 2013*). The Ire1α-Xbp1s pathway appears to be important for systemic UPR activation in vertebrates as well, since Xbp1s expression in murine POMC neurons activates Xbp1s in the liver, paralleling the improved hepatic insulin sensitivity and suppression of glucose production (*Williams et al., 2014*). Thus, although the secreted systemic factors that activate the UPR in distal tissues remain largely unknown, the UPR is a conserved mechanism that is activated by systemic factors to prepare the perceiving cells against stress by promoting proteostasis. Our work establishes that E2 is a systemic extrinsic regulator of UPR in adult HSCs that endows them with enhanced stress resistance, which is critical during regeneration.

## Materials and methods

### Mice

The mouse alleles used in this study were *Ubc-GFP* (C57BL/6-Tg(UBC-GFP)30Scha/J, JAX Stock #004353) (*Schaefer et al., 2001*), *Mx1-Cre* (C.Cg-Tg(Mx1-cre)1Cgn/J, JAX Stock #005673) (*Kuhn et al., 1995*), *Esr1$^{fl}$* (*Singh et al., 2009*), and *ERAI* (*Iwawaki et al., 2004*) on a C57BL/6 background. Male mice were used for most experiments unless otherwise noted. Mice were housed in AAALAC-accredited, specific-pathogen-free animal care facilities at Baylor College of Medicine (BCM), or University of Michigan (UM) with 12 hr light-dark cycle and received standard chow ad libitum. All procedures were approved by the BCM (protocol #AN-5858) or UM (protocol #PRO00007786) Institutional Animal Care and Use Committees.

### Drug and chemical treatments

Mice were injected subcutaneously with 100 µl of corn oil containing 2 µg estradiol (Sigma, St. Louis, MO) every day for 7 days unless otherwise noted. Oil treated mice and E2 treated mice were caged separately. Poly I:C (Amersham, Piscataway, NJ) was resuspended in PBS at 50 µg/ml, and mice were injected intraperitoneally with 0.5 µg/gram of body mass every other day for 6 days. To measure protein translation, mice were injected intraperitoneally with 500 µg of puromycin (Thermo-Fisher Scientific, Waltham, MA) and sacrificed one hour later. To promote HSC activation and mobilization, mice were intraperitoneally injected with 4 mg of cyclophosphamide (Bristol Myers Squibb, NY) on day −1, and then injected subcutaneously on successive days with 5 µg of human G-CSF (Amgen, Thousand Oaks, CA). Mice were sacrificed on day 3 or 6.

### Radiation studies

10- to 12-week-old male C57BL/6 mice were treated for 7 days with control oil or E2 followed by whole body irradiation at 600 cGy TBI using a cesium-137 irradiator. PB complete blood counts (CBC) were measured using a scil Vet abc Plus+ (scil Animal Care/Henry Schein Animal Health, Gurnee, IL) every other day. For assessment of hematopoietic content, 10- to 12-week-old male C57BL/6 mice were treated for 7 days with control oil or E2 followed by whole body irradiation at 600 cGy TBI and were euthanized at one or two weeks post-irradiation. For survival studies, 10- to 12-week-old C57BL/6 mice were irradiated with 800 cGy TBI after E2 regimen. An IACUC-approved protocol (AN-5858) was followed to monitor the survival of irradiated mice.

## Transplantation

10–12 week-old male C57BL/6 mice were randomly assigned to groups after receiving two doses of radiation (500 cGy) administered at least three hours apart (total 1000 cGy). 100 GFP$^+$ HSCs were co-transplanted with $2 \times 10^5$ GFP$^-$ WBM cells into each recipient (total volume of 100 ul; 1X HBSS, 2% heat-inactivated bovine serum). Injections were administered via retroorbial injection. For secondary transplantation, similarly irradiated recipients were injected with 2 million WBM cells from primary recipients (sacrificed at 16 weeks post-transplantation).

## Flow-cytometry and HSC isolation

Bone marrow cells were either flushed from the long bones (tibias and femurs) or isolated by crushing the long bones (tibias and femurs), pelvic bones, and vertebrae with mortar and pestle in Hank's buffered salt solution (HBSS) without calcium and magnesium, supplemented with 2% heat-inactivated bovine serum (GIBCO, Grand Island, NY). Cells were triturated and filtered through a nylon screen (100 µm, Sefar America, Kansas City, MO) or a 40 µm cell strainer (Fisher Scientific, Pittsburg, PA) to obtain a single-cell suspension. For isolation of CD150$^+$CD48$^{-/low}$Lineage$^-$Sca-1$^+$c-kit$^+$ HSCs (*Kiel et al., 2005*; *Oguro et al., 2013*), bone marrow cells were incubated with PE-Cy5-conjugated anti-CD150 (TC15-12F12.2; BioLegend, San Diego, CA), PE-conjugated (HM48-1; BioLegend) or PE-Cy7-conjugated anti-CD48, APC-conjugated anti-Sca-1 (Ly6A/E; E13-6.7), and biotin-conjugated anti-c-kit (2B8) antibody, in addition to antibodies against the following FITC-conjugated lineage markers: CD41 (MWReg30; BD Biosciences, San Jose, CA), Ter119, B220 (6B2), Gr-1 (8C5), CD2 (RM2-5), CD3 (KT31.1), and CD8 (53–6.7). For the isolation of GFP$^+$ HSCs, the FITC channel was left open and PE-conjugated lineage markers were used instead. For isolation of CD34$^-$CD16/32$^-$Lineage$^-$Sca-1$^-$c-kit$^+$ MEPs, CD34$^+$CD16/32$^-$Lineage$^-$Sca-1$^-$c-kit$^+$ CMPs, and CD34$^+$CD16/32$^+$Lineage$^-$Sca-1$^-$c-kit$^+$ GMPs, bone marrow cells were incubated with FITC-conjugated anti-CD34 (RAM34; eBiosciences, San Diego, CA), PE-Cy7 conjugated anti-CD16/32 (93; Biolegend), PE-Cy5-conjugated anti-Sca-1 (Ly6A/E; E13-6.7), and biotin-conjugated anti-c-kit (2B8) antibody, in addition to antibodies against the following PE-conjugated lineage markers: Ter119, B220 (6B2), Gr-1 (8C5), Mac-1 (M1/70), CD2 (RM2-5), CD3 (KT31.1), and CD8 (53–6.7). Unless otherwise noted, antibodies were obtained from BioLegend, BD Biosciences, or eBioscience (San Diego, CA). Biotin-conjugated antibodies were visualized using streptavidin-conjugated APC-Cy7. HSCs were sometimes pre-enriched by selecting c-kit$^+$ cells using paramagnetic microbeads and autoMACS (Miltenyi Biotec, Auburn, CA). Nonviable cells were excluded from sorts and analyses using the viability dye 4',6-diamidino-2-phenylindole (DAPI) (1 µg/ml). To analyze hematopoietic lineage composition, bone marrow cells or splenocytes were incubated with PerCPCy5.5-conjugated anti-B220, PE-conjugated anti-Ter119, APC-conjugated anti-CD3, APC-eFluor780-conjugated anti-Mac-1 (M1/70), and PE-Cy7-conjugated anti-Gr-1 antibodies. Annexin V staining was performed using Annexin V APC (BD Biosciences). For analysis of aggregated proteins, hematopoietic cells were stained with anti-c-kit, Sca-1, and lineage antibody mix and then fixed in Cytofix/Cytoperm Fixation and Permeabilization Solution (BD Biosciences) for 30 min. Fixed cells were then stained with the ProteoStat dye (1:10,000 in permeabilization solution) at 25°C for 30 min (Enzo Life Sciences, Farmingdale, NY). To measure protein translation in HSCs, bone marrow cells were stained for HSC markers, and then fixed/permeabilized using BD Cytofix/Cytoperm solution. Cells were incubated with anti-puromycin antibody (12D10, Millipore, Billerica, MA) diluted in BD Perm/Wash Buffer (BD Biosciences) at 1:200 for one hour at room temperature. Cells were washed once and incubated with Alexa Fluor 633-conjugated anti-mouse IgG2a antibody (ThermoFisher Scientific, Waltham, MA) for 30 min at room temperature, washed once, and stained with DAPI before analyzing by flow cytometry. Flow cytometry was performed with FACSAria II, FACSCanto II, LSR II, or LSRFortessa flow-cytometers (BD Biosciences).

## Quantitative real-time (reverse transcription) PCR

HSCs and other hematopoietic cells were sorted into Trizol (Life Technologies, Carlsbad, CA) and RNA was isolated according to the manufacturer's instructions. cDNA was made with random primers and SuperScript III reverse transcriptase (ThermoFisher Scientific, Waltham, MA). Quantitative PCR was performed using a ViiA7 Real-Time PCR System (ThermoFisher Scientific, Waltham, MA). Each sample was normalized to β-actin. Data were analyzed using the $2^{-\Delta\Delta ct}$ method. Primers to quantify cDNA levels are listed in *Supplementary file 1*. Analysis of spliced:unspliced XBP1 ratios

were conducted with the Taqman probes spliced XBP1 5' 6-FAM/TGCTGAGTC/ZEN/CGCAG-CAGGTGCA/IABkFQ-3' and unspliced XBP1 5'-HEX/CGCAGCACT/ZEN/CAGACTATGTGCACC/IABkFQ-3'. As an endogenous control, the Gapdh primer and probe assay (Applied Biosystems, ID no. Mm99999915_g1) was used. cDNA was amplified using TaqMan Universal PCR Master Mix Kit (ThermoFisher Scientific, Waltham, MA) according to the manufacturer's instructions.

## RNA-seq

HSCs were sorted into Trizol and RNA purified according to the manufacturer's instructions. DNase I treated RNA samples were amplified using the NuGEN Ovation RNA-Seq System (NuGEN, San Carlos, CA), and cDNA fragmented using a Covaris Ultrasonicator (Covaris, Woburn, MA). Fragmented cDNA was assembled into a sequencing library using KAPA Library Preparation Kit for Ion Torrent (KAPA Biosystems, Wilmington, MA) and sequenced on an Ion Proton Sequencer (ThermoFisher Scientific, Waltham, MA). Reads obtained from Ion Torrent sequencing were mapped to mm10 using STAR (version 2.5.2b [*Dobin and Gingeras, 2016*]), which was followed by differential expression analysis using DESeq2 (version 1.12.4 [*Love et al., 2014*]). Pairwise GSEA (GSEA, version 2.2.1 [*Subramanian et al., 2005*]) was performed, combinatorially, on the lists of differentially expressed genes derived from each comparison.

## ATAC-seq

ATAC-seq library was performed essentially as described in *Buenrostro et al. (2013)*, with some modifications. 5000 HSCs were sorted and incubated in 100ul lysis buffer (10 mM Tris-Cl, pH 7.4, 10 mM NaCl, 3 mM MgCl2 and 0.1% NP-40) for 3 min on ice. Immediately after lysis, the lysis buffer was removed by spinning at 500 x g for 5 min with 4°C table-top centrifuge. The nuclei pellet was then resuspended in 10μl of transposase reaction mix (2μl 5xTAPS buffer (50mM TAPS-NaOH, 25mM $MgCl_2$ (pH8.5)), 0.5μl Tn5 (0.885 mg/mL), 7.5μl of nuclear free water). The transposase reaction was incubated at 37°C for 30 min, and then stopped by adding 5μl of 0.2% SDS for 5 min at room temperature. The transposase reaction was then purified by a ZYMO DNA purification kit. Tagmentated DNA was eluted in 10μl nuclease free water, and a half of the elution volume was then used for PCR enrichment reaction with NEBNext Q5 Hot Start HiFi PCR Master Mix (NEB M0543S). The library amplification reaction included 25μl of 2x Q5 master mix, 5 μl of tagmented DNA template, 2.5μl of 25uM PCR primer 1 (final conc. 1.25μM), 2.5μl of 25μM barcoded PCR primer (final conc. 1.25μM) (all primers from SIGMA), and 15μl of nuclear free water. PCR protocol was 5 min at 72°C, 30 sec at 98°C, and then n cycles of 10 sec at 98°C, 30 sec at 63°C, 1 min at 72°C, where "n" was determined by qPCR with SYBRGreen to reduce GC and size bias. After that, the library was purified using SPRI beads and eluted in 20μl of nuclear free water (the final concentration ~ 30nM). Libraries were sequenced on the Illumina HiSeq 2000 platform. Reads were mapped to mm10 using Bowtie2 (Version: 2.2.6), and peaks were called on each sample individually using MACS2 (Version: 2.1.0.20151222). Differential peaks were obtained on overlapping peaks using PePr (Version: 1.1.18). Heatmaps were generated using deepTools computeMatrix algorithm whereby the center each location (from BED file) was used as the reference point and was extended ± 5kb from the center point.

## HSC culture and differentiation assays

For CFU assays, single freshly isolated HSCs from control and E2-treated *Esr*^fl/fl^and *Esr*^fl/fl^; *Mx1-Cre* mice were plated into 96-well plates containing M3434 MethoCult methylcellulose media (StemCell Technologies, Cambridge, MA). Colonies were collected, washed, and cytospun onto glass slides followed by Wright-Giemsa staining. For megakaryocyte differentiation assays 24 single HSCs were sorted directly into serum free, phenol red free medium (X-Vivo 15, Lonza, Allendale, NJ) supplemented with 50 ng/ml of SCF and TPO, and 10 ng/mL IL-3 and IL-6 (all from Peprotech, Rocky Hill, NJ), and cultured for three days. Following brief culture, the contents of all wells were combined and plated on collagen-based MegaCult media (StemCell Technologies, Cambridge, MA) in duplicate. 10 days after plating, collagen slides were dehydrated and stained for cholinesterase activity per the manufacturer's instructions. Analysis of chemically-induced UPR was performed by sorting freshly isolated LSK cells (1 x $10^4$) into X-Vivo 15 supplemented with 1% FBS, 50ng/mL SCF and TPO, and 10ng/mL IL-3 and IL-6. Cells were incubated for 16 hours in 5% $CO_2$ with combinations of vehicle (ethanol, final concentration 0.1%), E2 (200 μM), DMSO (final concentration 0.001%),

tunicamycin (0.6 µg/ml in DMSO), and thapsigargin (4 nM in DMSO). Cells were collected, washed, and plated on methylcellulose media for CFU assays.

## Immunohistochemistry

Femurs from control and E2-treated mice were fixed overnight in 4% paraformaldehyde and then decalcified in 10% EDTA for 3-4 days, embedded in 8% gelatin to prepare 6 µm sections. Slides were incubated with rat anti-CD41 antibody (eBioscience, San Diego, CA) diluted 1:500 in blocking buffer (4% goat serum, 0.4% BSA, 0.1% NP-40 in PBS) overnight at 4 °C. Slides were rinsed and incubated with Alexa Fluor 488-conjugated goat anti-rat antibody (ThermoFisher Scientific, Waltham, MA) at 1:500 dilution in blocking buffer containing DAPI. Images were obtained using a Leica DMI 6000B microscope.

## Immunoblotting

Immunoblotting was performed as described previously (Nakada et al., 2010). Briefly, the same number of cells (20,000-30,000) from each test population were sorted into Trichloroacetic acid (TCA) and adjusted to a final concentration of 10% TCA. Extracts were incubated on ice for 15 minutes and spun down for 10 minutes at 13,000 rpm at 4°C. The supernatant was removed and the pellets were washed with acetone twice then dried. The protein pellets were solubilized with Solubilization buffer (9 M Urea, 2% Triton X-100, 1% DTT) before adding LDS loading buffer (Invitrogen, Carlsbad, CA). Proteins were separated on a Bis-Tris polyacrylamide gel (ThermoFisher Scientific, Waltham, MA) and transferred to a PVDF membrane (Millipore, Billerica, MA). Antibodies were anti-Ire1α (Cell Signaling Technology, 3294), anti-Xbp1s (Biolegend, Poly6195), and anti-ß-actin (A1978, Sigma).

## Cell lines

293T cells were obtained from ATCC (CRL-3216) and were tested negative for mycoplasma contamination.

## Luciferase assay

Approximately 4kb upstream of the Ire1α TSS was cloned into pGL2 Basic. ERα was expressed from pcDNA-HA-ER WT (Addgene #49498). 293T cells were transfected with the indicated combination of the plasmids in OptiMEM (Invitrogen, Carlsbad, CA). 6 hours after transfection, the media was replaced with phenol red free X-vivo15 media supplemented with 10% charcoal-stripped FBS (Invitrogen, Carlsbad, CA). Cells were treated with or without $10^{-7}$M of E2 dissolved in ethanol for 20 hours before cells were harvested for luciferase activity measurement performed according to the manufacturer's instruction (Promega, Madison, WI).

## ChIP assays

Cells were fixed with 1% paraformaldehyde for 10 minutes at room temperature. 125 mM glycine was then added and incubated at room temperature for 5 minutes. Cells were collected at 500 x g for 10 minutes at 4°C and washed twice with PBS. The pellet was lysed in Nuclear Lysis Buffer (10 mM Tris-HCl, pH 7.5, 1 mM EDTA, 1% SDS, 1x protease inhibitors (Roche) for 20 minutes on ice. Samples were then diluted with PBS to bring SDS concentration to 0.25% and sonicated in a Bioruptor for 23 minutes (settings: High, 30 seconds on and 30 seconds off). Lysates were diluted 1:1.5X with equilibration buffer (10mM Tris, 233 mM NaCl, 1.66 % TritonX-100, 0.166 % sodium deoxycholate, 1 mM EDTA, inhibitors) and centrifuged at 14,000 x g, 4°C for 10 minutes. Supernatant was transferred to a new tube (10% of the sample was saved as input). ERa (ab32063, Abcam, Cambridge, MA) and IgG antibodies were added to the lysate and incubated at 4°C overnight. Protein A/G Dynabeads (ThermoFisher Scientific) were added to the chromatin and incubated for 2 hours at 4°C. Beads were washed subsequently with RIPA-LS (10 mM Tris-HCl/pH 8.0, 140 mM NaCl, 0.1% SDS, 1% Triton X-100, 1 mM EDTA, 0.1% sodium deoxycholate) for 3 times, RIPA-HS (10 mM Tris-HCl/pH 8.0, 500 mM NaCl, 0.1% SDS, 1% Triton X-100, 1 mM EDTA, 0.1% sodium deoxycholate) for 3 times, RIPA-LiCl (10 mM Tris-HCl/pH 8.0, 1 mM EDTA, LiCl 250mM, 0.5% NP-40, 0.5% sodium deoxycholate) for 3 times, and 10 mM Tris-HCl, pH 8.0 for 3 times, after of which samples were transferred to a new tube. Beads were then incubated with 50 µl elution buffer (0.5% SDS, 300 mM

NaCl, 5 mM EDTA, 10 mM Tris HCl pH 8.0) containing 2 µl of 10 mg/ml Proteinase K for 1 hour at 55°C and 8 hours at 65°C to reverse crosslink, and supernatant was transferred to a new tube. Another 20 µl of elution buffer and 1 µl of Proteinase K were added to the beads and incubated for 1h at 55°C, and the supernatants were combined. Finally, DNA was purified with SPRI beads and used for qPCR with primers listed in Supplementary File 1.

### CRISPR/Cas9-mediated genome editing

Genome editing was performed essentially as described previously (*Gundry et al., 2016*). Briefly, protospacer sequences for each target gene were identified using the CRISPRscan algorithm (www.crisprscan.org). DNA templates for sgRNAs were made in a PCR reaction using a forward primer containing a T7 promoter, a reverse primer specific for the 3' end of the improved scaffold, and pKLV-U6gRNA-EF(BbsI)-PGKpuro2ABFP plasmid (Addgene # 62348). c-kit$^+$ or LSK cells were isolated by MACS or FACS, respectively, and cultured in X-Vivo 15 (Lonza, Allendale, NJ) supplemented with 50 ng/ml of SCF and TPO, and 10 ng/mL IL-3 and IL-6 (all from Peprotech, Rocky Hill, NJ) for 1 hour. $1x10^4$ to $1x10^5$ c-kit$^+$ cells or $5x10^4$ LSK cells were resuspended into 10 µl of Buffer T (Invitrogen), mixed with 1 µg sgRNA and 1 µg Cas9 protein (PNA Bio), and electroporated using a Neon transfection system (ThermoFisher Scientific) using the following parameters; 1700V, 20ms, 1 pulse. After overnight culture in X-Vivo media, live cells were sorted and used for subsequent experiments. For clonal analysis of editing efficiency, single LSK cells were sorted into M3434 MethoCult methylcellulose media (StemCell Technologies, Cambridge, MA) after electroporation. Colonies were collected 12 days later, and purified genomic DNA was sequenced via Sanger sequencing. Indels were quantified using the TIDE deconstruction algorithm (*Brinkman et al., 2014*). See Supplementary File 1 for oligo sequence.

### Quantification and statistical analysis

Statistical analyses were performed using GraphPad Prism 5.0c (GraphPad Software Inc., San Diego, CA). To ensure the reproducibility of our findings, all data were derived from multiple independent experiments performed on different days. Sample sizes were chosen based on observed effect sizes and standard errors from previous experiments, and data was checked for normality and similar variance between groups. Only males were used in these studies, and all experimental units were age-matched. In the case of investigating ERα deletion, untreated floxed littermates were used as controls. Animal studies were performed without blinding. To test statistical significance of the means of two samples, two-tailed Student's t tests or Mann Whitney U Tests were performed. Statistical significance of multiple groups was evaluated with a two-way ANOVA or repeated measures ANOVA followed by Bonferroni's test for multiple comparisons. Analysis of three or more groups with unequal sample sizes was conducted using the Kruskal-Wallis H Test followed by Dunn's multiple comparisons test. Survival curves were tested for statistical significance with a log-rank test. A p-value of less than 0.05 was considered significant. Values are reported as mean ± SD unless stated otherwise.

### Data and software availability

The accession number for the RNA sequencing data reported in this paper is GSE99120. The accession number for the ATAC-seq data is SRP12895.

## Acknowledgements

This work was supported by the National Institutes of Health (CA193235 and DK107413 to DN and HL132392 to QL). RHC was supported by NIH T32 DK060445. The authors declare no competing financial interests. We thank Dr. Masayuki Miura (University of Tokyo) for providing the ERAI strain, and Catherine Gillespie for critical reading of the manuscript.

## Additional information

### Funding

| Funder | Grant reference number | Author |
|---|---|---|
| National Institutes of Health | T32 DK060445 | Richard H Chapple |
| National Heart, Lung, and Blood Institute | HL132392 | Qing Li |
| National Cancer Institute | CA193235 | Daisuke Nakada |
| National Institute of Diabetes and Digestive and Kidney Diseases | DK107413 | Daisuke Nakada |

The funders had no role in study design, data collection and interpretation, or the decision to submit the work for publication.

### Author contributions
Richard H Chapple, Data curation, Formal analysis, Investigation, Writing—original draft, Writing—review and editing; Tianyuan Hu, Formal analysis, Investigation, Writing—review and editing; Yu-Jung Tseng, Lu Liu, Ayumi Kitano, Victor Luu, Kevin A Hoegenauer, Formal analysis, Investigation; Takao Iwawaki, Resources; Qing Li, Formal analysis, Supervision, Funding acquisition, Investigation; Daisuke Nakada, Conceptualization, Supervision, Funding acquisition, Writing—original draft, Project administration, Writing—review and editing

### Author ORCIDs
Richard H Chapple (iD) http://orcid.org/0000-0002-0375-2043
Daisuke Nakada (iD) http://orcid.org/0000-0001-6010-7094

### Ethics
Animal experimentation: Mice were housed in AAALAC-accredited, specific-pathogen-free animal care facilities at Baylor College of Medicine (BCM), or University of Michigan (UM) with 12hr light-dark cycle and received standard chow ad libitum. All procedures were approved by the BCM or UM Institutional Animal Care and Use Committees (protocol #AN-5858).

### Decision letter and Author response
Decision letter https://doi.org/10.7554/eLife.31159.022
Author response https://doi.org/10.7554/eLife.31159.023

## Additional files

### Supplementary files
• Supplementary file 1. List of primers used for quantitative real-time PCR, ChIP-seq, and sgRNA systhesis. All primers are listed in the 5' to 3' direction.
DOI: https://doi.org/10.7554/eLife.31159.015
• Transparent reporting form
DOI: https://doi.org/10.7554/eLife.31159.016

### Major datasets
The following datasets were generated:

| Author(s) | Year | Dataset title | Dataset URL | Database, license, and accessibility information |
|---|---|---|---|---|
| Daisuke Nakada, Richard H Chapple, Kevin A Hoegenauer | 2017 | ATAC-Seq of hematopoietic stem cells from vehicle and estrogen-treated mice | https://www.ncbi.nlm.nih.gov/sra/SRP128950 | Publicly available at NCBI BioProject (Accession no: SRP128950) |
| Nakada D, Chapple R, Hoegenauer KA | 2017 | RNA-Seq of hematopoietic stem cells from vehicle and estrogen-treated mice | https://www.ncbi.nlm.nih.gov/geo/query/acc.cgi?acc=GSE99120 | Publicly available at the NCBI Gene Expression Omnibus (accession no: GSE99120). |

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
