## [Decision Letter]

Thank you for sending your article entitled "ERα promotes hematopoietic regeneration through the Ire1α-mediated unfolded protein response" for peer review at *eLife*. Your article has been evaluated by Marianne Bronner (Senior Editor) and three reviewers, one of whom is a member of our Board of Reviewing Editors.

Main concerns raised relate to the link between ER, IRE1 and target genes such as Myc.

1) Regarding knocking down IRE1, please see reviewer 2 point 2 and reviewer 3 point 4.

2) Regarding the link between ER and IRE1, please see reviewer 1 point 3 and reviewer 3 point 3.

3) Regarding the effect on target genes, including Myc, please see reviewer 1 point 2 and reviewer 2 point 2.

Moreover, there was agreement on the need to further clarify the cell autonomous/non-autonomous action of E2, including a deeper discussion of the consistency of results (see reviewer 1 point 1 and reviewer 3 point 1.

Addressing the other points raised would of course be beneficial; however there is a concern that addressing the two major concerns may require mouse models or time course experiments that might require a longer timeframe of revision than normally allowed by *eLife*.

*Reviewer #1:*

1) The authors investigate in detail the role and mechanism of action of estrogen stimulation on hematopoietic stem cells, and perform important transplantation experiments that were missed in previous studies. Linking the UPR to the E2 response is very novel and a finding that advances not only the HSC field.

The overall work is very good and elegant, however my major concern is the repeated comment that ER works cell intrinsically only. It certainly works within HSCs, however without any experiments with chimeric mice in which either the hematopoietic system or the stroma lack ERα it is impossible to conclude that ER signaling does not play a role indirectly through the stroma/ HSC niche too. In fact, some in vitro experiments only partially reproduce the in vivo findings, and in most experiments HSCs harvested from treated mice are used, rather than HSCs treated ex vivo. The authors could either generate the chimeras, which is however time-consuming as a pre-experiment, or be more careful in their comments.

2) The other major missing point in terms of functional characterization of HSCs from E2-treated mice is a limiting dilution transplantation assay, so that not only groups of 100 HSCs, but also smaller doses, e.g. single or 5-10 HSCs can be assessed and frequencies can be estimated. This seems important especially in view of up-regulation of cMyc following E2 treatment, with Myc activation being consistently linked to loss of HSC function. Is it possible that E2 treatment leads to further proliferation of already active HSCs rather than dormant (e.g. label retaining) ones?

3) Another major point regards the connection between ERα and the UPR. Is Ire1α a direct target of ERα? This could already be known, and therefore mentioned, or checked through bioinformatics analysis of the Ire1α promoter, and ideally a ChiP experiment.

4) Finally, I don't see anywhere the gating strategy used to identify HSC and progenitor cell populations. It is very important that these are shown as different gates are used by different research groups and it is easy to be transparent about the gates used, as long as they are consistent throughout the work shown.

5) The survival experiment described in Figure 2 does not really add much to the previous experiment, showing recovery from a lower dose of irradiation. I would either remove the experiment entirely to avoid any reference to letting mice die as an experimental end point, which is unethical, or explain how the mice were humanely monitored.

Reviewer #2:

In their manuscript "ERα promotes hematopoietic regeneration through the IRE1α-mediated unfolded protein response" Nakada et al. provide evidence that estrogen receptor (ER) signaling promotes HSC regeneration following irradiation at least in part due to activation of the Ire1a branch of the unfolded protein response (UPR). Altogether the hypothesis is interesting and the work is a nice extension of prior studies in this area.

I have a few major comments relating to improving the quality of the study:

1) The implication of the data (and prior work from the Morrison lab) is that ER signaling promotes HSC proliferation. As such it would be important to distinguish whether the mechanism by which ER signaling promotes recovery is due to protection from apoptosis (given Annexin V data), proliferation or both, as either mechanism can lead to faster recovery of BM cell numbers.

2) Along these lines, is UPR directly responsible for increased Myc/E2F gene levels and cell proliferation or vice versa? This can be tested in regenerative conditions using the CRISPR system developed by the authors and would define the epistatic relationship between these important and functionally interleaved gene programs in hematopoietic regeneration.

3) Overall a key finding of the paper is the dependence on Ire1α based on CRISPR. To fully substantiate the Ire1α findings to the reader I recommend that the authors show the full timecourse of hematopoietic regeneration following IR such as shown in Figure 2, and demonstrate by directly interrogating the BM whether similar kinetics of BM and LSK recovery are observed as in Figure 3, at least looking in the donor compartment if not BM as a whole.

4) Figure 5 shows sensitization of LSK to stressors; on the flipside does Ire1α KD also abrogate any increases in colony forming activity in LSK, or is the colony forming phenotype only specific to HSCs?

*Reviewer #3:*

This study is an extension of a previous study showing E2 increases HSC proliferation and frequency and erythropoiesis (Nakada et al., 2014), showed E2 increases broad myelopoiesis, and uncovered the mechanistic link between E2 and UPR, a protective mechanism for HSCs (van Galen et al., 2014). They showed that IRE1-XBP1 branch of the UPR is responsible for E2's effect on HSCs, as deleting IRE1 using CRISPR in LSK cells blunted E2's effect on HSCs.

1) Non-autonomous UPR refers to the phenomenon that ER stress in one tissue induces UPR in a remote tissue. If the authors claim that what they found here represents an example of non-autonomous UPR, they would need to show that ER stress in ovaries leads to release of E2, which activates UPR in HSCs. However, it does not seem to be the case here.

2) The observation that E2 increases myelopoiesis but not lymphopoiesis is interesting. Do myeloid primed HSCs have increased ER stress than lymphoid primed HSCs? The authors propose IRE1 as an underlying mechanism. However, in Figure 5, it appears that the effects of E2 and IRE1α are uncoupled. E2 increased WBC but IRE1α deletion had no effect on WBC. E2 had no effect on RBC but IRE1α deletion reduced RBC.

3) The authors performed RNA-seq and found that UPR genes are upregulated in E2-treated HSCs. Importantly, they showed that the UPR upregulation upon E2 is dependent on ERα. How does E2-ERα upregulate IRE1? Does IRE1 promoter have ERα binding site? Does ERα bind to IRE1 promoter?

4) In Figure 5, what's the editing efficiency? Typically, a pool of CRISPR edited cells do not show knockdown, but need to be selected for single clones that have efficient knockdown. Since the novel aspect of this study is the connection of E2 with UPR and the knockout mouse models for all three branches of UPR are well established, it makes more sense to use the KO mice.

5) The effect of E2 on hematopoiesis is inconsistent throughout the manuscript. e.g. in Figure 2 significantly increases platelets. However, in Figure 5 has no effect on platelets. It is unclear how robust the effect of E2 is.

6) Figure 5—figure supplement 1 is missing.

[Editors' note: further revisions were requested prior to acceptance, as described below.]

Thank you for resubmitting your work entitled "ERα promotes murine hematopoietic regeneration through the Ire1α-mediated unfolded protein response" for further consideration at *eLife*. Your revised article has been favorably evaluated by Marianne Bronner (Senior Editor), and a Reviewing Editor.

The manuscript has been improved but there are some remaining issues that need to be addressed before acceptance, as outlined below:

1) Subsection “Estradiol promotes protective unfolded protein response in HSCs”, third paragraph. Please state clearly that clonal analysis of electroporated LSK cells indicated that about 25% of the clones were deleted (Figure 6—figure – supplement 1D). Nevertheless, bulk-modified cell populations used for subsequent experiments showed that….

2) Subsection “Estradiol activates the unfolded protein response (UPR) pathway in HSCs”, first paragraph. In line with the detailed explanation presented in the response letter, please modify the text to indicate that a number of Myc target genes were induced by E2, even though c-Myc, N-Myc and L-Myc were not. It would be helpful to include the comments provided in this section of the response letter within the Discussion section of the manuscript.

3) Please indicate at least in the Materials and methods that, based on Figure 1—figure – supplement 1A, HSCs were sorted as CD48-/low (and for brevity named CD48-) according to Kiel et al., Oguro et al. etc. There is much confusion in the field about where to place CD48 gates, and the more clarity is used in the presentations, the better. Overall, the gating schemes should include the experimental treatments and not just the control as presented in Figure 1—figure supplement 1. This is particularly true for readers who are evaluating how the E2 and IR treatment conditions change distribution of markers such as CD48 and CD150.

4) Please include the comments about variable effects on platelets production presented in the response letter within the Discussion section.

---

## [Author Response]

Main concerns raised relate to the link between ER, IRE1 and target genes such as Myc.1) Regarding knocking down IRE1, please see reviewer 2 point 2 and reviewer 3 point 4.

Regarding the efficiency of our CRISPR protocol, we have performed a clonal analysis of edited cells. We performed our CRISPR electroporation protocol outlined in our previously published manuscript (Gundry et al., 2016). Briefly, RNP complexes are assembled by co-incubating Cas9 protein and either 1 or 2 sgRNAs (Rosa26 and Ire1α respectively), and then delivered to LSKs via electroporation. We then sorted single LSKs into 96 well plates of methylcellulose medium to allow for colony formation. Clones were isolated and genomic DNA was profiled using Sanger sequencing. We then used a decomposition algorithm (TIDE) specifically designed to identify and quantify the major mutations at projected cut sites. The results of this analysis are highlighted in Figure 6—figure supplement 1. We found that a large percentage (>75%) of clones contained a mutation using this approach. Of those that were edited, one-third contained a large deletion consisting of an ablation of the DNA between the two sgRNA sites. The remaining colonies contained 1-4bp indels that were characterized as homozygous, heterozygous, or undetermined. Given the highly efficient editing of our technique, combined with our immunoblot results showing significant reduction in total Ire1α protein levels using this approach (Figure 6), we concluded that this technique is sufficient for use in investigating the role of Ire1α in HSPCs.

In regards to the issue of whether “bulk” edited cells without selection of edited cells could be used to study the function of genes, we note that recent studies have used a similar “bulk” approach as we did. For example, Michael Kharas and colleagues deleted METTL3 from leukemia cell lines without selecting clones, and observed that METTL3 deletion reduced proliferation and survival in the bulk cell population (Nature Medicine. 23:1369). Ken Anderson and colleagues deleted KDM6B from multiple myeloma cells and observed reduced growth and survival of these cells (Leukemia. 31:2661).

Additionally, CRISPR dropout screens that have brought important insights into various area of biology depends on the principle that “bulk” edited cells with heterogeneous mutation profiles, even including some unedited cells, could be used to observe phenotypic changes, such as survival or gene expression changes (for example, Cell. 163:1515, Science. 350:1096, Cell Reports. 17:1193). These screens are performed by selecting cells that have been transduced with sgRNA libraries, but not by selecting clones with verified knockout efficiencies (Discussed by David Root in Genome Biol. 16: 260.). It is inevitable that the efficiency of knockout based on the CRISPR/Cas9 system is the key for the success of such screens and other experiments that depend on the phenotypic changes in heterogeneously edited cell population. Our results indicate that we do achieve efficient knockout of Ire1α (Figure 6, Figure 6—figure supplement 1), and observe clear phenotypic changes upon deletion of Ire1α (Figure 6).

2) Regarding the link between ER and IRE1, please see reviewer 1 point 3 and reviewer 3 point 3.

Two reviewers brought forth inquiry about the regulation of Ire1α activation on HSPCs. We started by investigating the changes that occur to the global chromatin landscape in hematopoietic precursor cells using ATACseq. Using bioinformatics analyses, we identified differentially regulated peaks in Oil- and E2-treated HSCs, and discovered that differential ATAC-seq peaks in E2-treated HSCs are enriched for estrogen response elements (ERE). One of the E2-specific ATAC peak was found upstream of Ern1 (which encodes for Ire1α) and contained a consensus ERE. To determine if this region was sufficient to activate Ire1α expression, we cloned the upstream regulatory region of Ire1α into a luciferase dual reporter assay and found that luciferase activity increased in an E2- and ERα-dependent manner. This activity depended on ERE since mutating the ERE ablated the transactivation property of this sequence. Finally, we performed ChIP-qPCR to test whether ERα occupies Ire1α upstream locus in HSPCs. We observed an increase in ERα binding at the ERE locus of Ire1α in E2-treated hematopoietic progenitor cells. These results indicate that Ire1α is a direct target of ERα in HSPCs. These data have been compiled into a new section of the manuscript, the results of which can be found in the newly added Figure 5.

3) Regarding the effect on target genes, including Myc, please see reviewer 1 point 2 and reviewer 2 point 2.

We adopted our CRISPR strategy to address the potential relationship between Ire1 and Myc target activation. We cultured CRISPR-edited LSK cells in vitrowith or without exogenous E2 and checked mRNA expression of several Myc targets identified in the GSEA. We first confirmed that Ire1α is induced upon E2 treatment in vitro (Figure 4—figure supplement 1), similar to what we observed in vivo (Figure 4). E2 induced the expression of several Myc target genes identified by GSEA (Figure 4—figure supplement 1). Among these Myc target genes, some of them depended on Ire1α to become induced by E2 (Figure 4—figure supplement 1), whereas other genes did not (Figure 4—figure supplement 1). This indicates that some of these target genes are downstream of Ire1α, but the entire Myc target gene set does not behave in this manner. We also note that although GSEA detected global upregulation of Myc target genes, c-Myc, L-Myc, or N-Myc were not induced. Since Myc has a large impact on the expression of genes involved in cell growth and proliferation (Cell. 151:56-67), it is likely that many stimuli that promote proliferation will exhibit increased “Myc target” expression by GSEA even though those stimuli may not involve Myc.

An additional question was on the heterogeneity of HSCs, and whether E2 is acting upon a certain subset of HSCs. We addressed this question by using CD150 as a surrogate marker for myeloid-biased HSCs (PNAS. 107:5465,

Cell Stem Cell. 6:265). We sorted CD150^HI^, CD150^LO^, and CD150^NEG^ CD48^-^ LSKs from Oil- and E2-treated mice and measured Ire1α expression. Ire1α expression was increased upon E2-treatment equally in each population, suggesting that E2 is not specifically acting on myeloid-biased CD150^HI^ HSCs (Figure 4—figure supplement 2).

These data address the main concerns brought forth by the reviewers.

Moreover, there was agreement on the need to further clarify the cell autonomous/non-autonomous action of E2, including a deeper discussion of the consistency of results (see reviewer 1 point 1 and reviewer 3 point 1.

In addition to the aforementioned data, we revised the manuscript to address several other comments. Of note, there was discussion regarding the cell autonomous vs. cell non-autonomous action of E2 on HSPCs and an issue of clarifying the consistency of the results. We have revised the text to reflect our position that E2 is acting as a systemically acting extrinsic regulator of HSC function, and removed any claims implying that ERα acts solely on HSCs. Moreover, we performed additional experimental shown in Figure 6 and examined how E2 and Ire1α regulates HSPCs recovery in the bone marrow.

These additional experiments highlight the robustness of E2 as a positive regulator of hematopoietic regeneration. We believe that the new data and revisions have resulted in substantial improvement of the original manuscript.

Addressing the other points raised would of course be beneficial; however there is a concern that addressing the two major concerns may require mouse models or time course experiments that might require a longer timeframe of revision than normally allowed by eLife and it is for this reason that we invite you to let us know what timeline you could propose.

Reviewer #1:

1) The authors investigate in detail the role and mechanism of action of estrogen stimulation on hematopoietic stem cells, and perform important transplantation experiments that were missed in previous studies. Linking the UPR to the E2 response is very novel and a finding that advances not only the HSC field.The overall work is very good and elegant, however my major concern is the repeated comment that ER works cell intrinsically only. It certainly works within HSCs, however without any experiments with chimeric mice in which either the hematopoietic system or the stroma lack ERα it is impossible to conclude that ER signaling does not play a role indirectly through the stroma/ HSC niche too. In fact, some in vitro experiments only partially reproduce the in vivo findings, and in most experiments HSCs harvested from treated mice are used, rather than HSCs treated ex vivo. The authors could either generate the chimeras, which is however time-consuming as a pre-experiment, or be more careful in their comments.

We agree that we cannot exclude the possibility that E2 affects the HSC niche to improve hematopoietic regeneration. This could potentially explain why we only see a partial rescue of hematopoietic cells after irradiation (Figure 2). We have toned down our comments that the effect of E2 is solely dependent on ERα in hematopoietic cells (subsection “Estradiol promotes the regenerative capacity of HSCs”, third and last paragraphs; subsection “Estradiol promotes hematopoietic recovery after irradiation”, first paragraph; subsection “Estradiol accelerates hematopoietic progenitor cell regeneration after ionizing radiation”, last paragraph). Additionally, we revised the Discussion to suggest that the HSC niche could also be a contributing factor to the increased hematopoietic regeneration (third paragraph).

2) The other major missing point in terms of functional characterization of HSCs from E2-treated mice is a limiting dilution transplantation assay, so that not only groups of 100 HSCs, but also smaller doses, e.g. single or 5-10 HSCs can be assessed and frequencies can be estimated. This seems important especially in view of up-regulation of cMyc following E2 treatment, with Myc activation being consistently linked to loss of HSC function. Is it possible that E2 treatment leads to further proliferation of already active HSCs rather than dormant (e.g. label retaining) ones?

In terms of the effect of E2 on HSC proliferation, we have previously published the results of an H2B-GFP labeling experiment (Nature. 505:555-558). We demonstrated that E2 promotes proliferation of both H2BGFP(high) and H2B-GFP(low) HSCs contains dormant and active HSCs, respectively, suggesting that E2 is not selective on the subtypes of HSCs it acts on.

To further test whether E2 acts on a particular subpopulation of HSCs, we investigated Ire1α expression HSC subpopulations distinguished by the level of CD150 expression. It has been shown by Derrick Rossi’s group that CD150-high HSCs are enriched for myeloid-biased HSCs, whereas HSCs with lower CD150 expression contain balanced and lymphoid-biased HSCs (PNAS. 107:5465). It is also known that CD150-high HSCs are enriched for those with lower side population (SP) phenotype, which are also more quiescent than the upper SP HSCs (Cell Stem Cell. 6:265). Ire1α expression did not differ between these populations, suggesting that E2 does not specifically act on myeloid-biased CD150-high HSCs. We have included these data in Figure 4—figure supplement 2.

We also note that although GSEA detected global upregulation of Myc target genes, c-Myc, L-Myc, or N-Myc were not induced. Since Myc has a large impact on the expression of genes involved in cell growth and proliferation (Cell. 151:56-67), it is likely that many stimuli that promote proliferation will exhibit increased “Myc target” expression by GSEA even though those stimuli may not involve Myc. Additionally, it has been shown that adult HSCs are generally c-Myc-low, but fetal liver HSCs, which are more proliferative than adult HSCs and have extensive self-renewal capacity, have higher Myc expression than adult HSCs (Nature Immunology. 11:207). Interestingly, we found that the expression of Lin28 and Hmga2 are induced in E2-treated adult HSCs. These two genes are highly expressed in fetal liver HSCs compared to adult HSCs and regulate self-renewal (Nature Cell Biology. 15:916, J Exp Med. 213:1497). This raises a possibility that E2 induces a fetal-type self-renewal program in adult HSCs (potentially through Lin28 or Hmga2), paralleling the activation of the cell cycle and increased myeloid reconstitution potential. We hope to test this hypothesis in near future.

3) Another major point regards the connection between ERα and the UPR. Is Ire1α a direct target of ERα? This could already be known, and therefore mentioned, or checked through bioinformatics analysis of the Ire1α promoter, and ideally a ChiP experiment.

For the full response, please see response to main concern point #2 above. The revised manuscript includes a new section in which we demonstrate that ERα associates with an upstream locus of Ire1α, highlighting that Ire1α is a direct target of ERα. We used ATAC-Seq to identify differentially regulated regions of accessible chromatin in HSPCs after E2 treatment, and identified a differentially regulated region of the Ire1α upstream region that contained a consensus estrogen response element (ERE). Using a dual luciferase assay and ChIP-qPCR assay, we confirmed that this region possesses E2- and ERα-dependent transactivation property, and that ERα occupies this site upon E2 stimulation in HSPCs. These new data are provided in the new Figure 5.

4) Finally, I don't see anywhere the gating strategy used to identify HSC and progenitor cell populations. It is very important that these are shown as different gates are used by different research groups and it is easy to be transparent about the gates used, as long as they are consistent throughout the work shown.

We have included the gating strategy for HSCs in Figure 1—figure supplement 1.

5) The survival experiment described in Figure 2 does not really add much to the previous experiment, showing recovery from a lower dose of irradiation. I would either remove the experiment entirely to avoid any reference to letting mice die as an experimental end point, which is unethical, or explain how the mice were humanely monitored.

We included these experiments because it has been previously established that the main cause of mortality after radiation is due to hematopoietic failure. The survival analysis is a standard assay used in many of the papers we cited in the manuscript that describes factors regulating hematopoietic regeneration after irradiation injury (for example, Nature Medicine. 23, 91–99). We monitored morbidity in accordance to IACUC guidelines for humane endpoints for laboratory animals, and this will be highlighted in the revised text.

Reviewer #2:

In their manuscript "ERα promotes hematopoietic regeneration through the IRE1α-mediated unfolded protein response" Nakada et al. provide evidence that estrogen receptor (ER) signaling promotes HSC regeneration following irradiation at least in part due to activation of the Ire1a branch of the unfolded protein response (UPR). Altogether the hypothesis is interesting and the work is a nice extension of prior studies in this area.I have a few major comments relating to improving the quality of the study:1) The implication of the data (and prior work from the Morrison lab) is that ER signaling promotes HSC proliferation. As such it would be important to distinguish whether the mechanism by which ER signaling promotes recovery is due to protection from apoptosis (given Annexin V data), proliferation or both, as either mechanism can lead to faster recovery of BM cell numbers.

We agree that both protection from apoptosis and increased proliferation may contribute of the faster recovery of BM cells. Determining the relative contribution of these two processes on BM recovery is a challenge, since we will need to block apoptosis and proliferation of HSPCs separately and examine the consequences of blocking these processes on BM recovery after irradiation. A previous report demonstrated that overexpression of BCl^-^2 substantially blocked apoptosis of BM cells after irradiation, but the BM cell number nonetheless dropped to ~10% of the steady-state level (Blood. 91:2272). Blocking HSC proliferation is also difficult since multiple cyclin/CDKs act redundantly (reviewed in J Cell Biol. 195:709). We modified the text to conclude that E2 promotes HSC proliferation and block IR-induced apoptosis, both of which may contribute to the faster recovery of BM cells, and avoid potential over-interpretation regarding the relative importance of preventing apoptosis vs. promoting proliferation.

2) Along these lines, is UPR directly responsible for increased Myc/E2F gene levels and cell proliferation or vice versa? This can be tested in regenerative conditions using the CRISPR system developed by the authors and would define the epistatic relationship between these important and functionally interleaved gene programs in hematopoietic regeneration.

For the full response to this comment, please main concern point #3 above. We now provide new data in Figure 4—figure supplement 1 showing that some Myc target genes are dependent on Ire1α for the induction after E2 treatment, whereas some other targets do not depend on Ire1α. It is likely that the Myc targets identified in the GSEA are not collectively downstream of Ire1α, and that other Ire1α- or UPR-independent mechanisms are likely to be in play.

3) Overall a key finding of the paper is the dependence on Ire1α based on CRISPR. To fully substantiate the Ire1α findings to the reader I recommend that the authors show the full timecourse of hematopoietic regeneration following IR such as shown in Figure 2, and demonstrate by directly interrogating the BM whether similar kinetics of BM and LSK recovery are observed as in Figure 3, at least looking in the donor compartment if not BM as a whole.

We thank the reviewer for this advice. We have performed the analysis of quantifying HSPCs in the bone marrow after transplanting Ire1αedited LSK cells with or without E2 treatment. Whereas E2 treatment increased the frequency of Rosa26-edited LSK cells and myeloid progenitor cells, E2 treatment failed to promote HSPC regeneration when Ire1α was deleted. These results reinforce the idea that E2 promotes hematopoietic regeneration in a manner dependent on Ire1α. These new data are shown in Figure 6. We opted not to perform detailed time course experiment after Rosa26- or Ire1α-editing and transplantation, followed by oil- or E2-treatment, since not only does this experiment use large numbers of mice (please note, that the same recipient mouse cannot be bled on two consecutive time points particularly after irradiation since it increases the mortality rate, requiring multiple cohorts for bleeding at different time points. Also LSK cells are scant before day 14 after TBI) but also because analyzing HSPCs as suggested by this reviewer is a more direct way to examine hematopoietic regeneration. In fact, we observed a much more clear effect of E2 and Ire1α deletion when we analyzed HSPCs as suggested.

4) Figure 5 shows sensitization of LSK to stressors; on the flipside does Ire1α KD also abrogate any increases in colony forming activity in LSK, or is the colony forming phenotype only specific to HSCs?

We performed a CFU assay with LSK cells isolated from oil- or E2-treated mice, similar to what we performed for HSCs (Figure 1). We did not see significant effects of E2 on the frequency or the types of colonies formed after E2 treatment when LSK cells were plated, suggesting that the effects of E2 in enhancing the ability of form immature myeloid colonies are specific to HSCs.

Also, as shown in our previous Figure 5 (now Figure 6), left panel black bars, deletion of Ire1α in control LSK cells showed no difference in CFU potential. Thus, Ire1α appears to be more important for colony formation when cells are challenged with proteotoxic stress.

Reviewer #3:

This study is an extension of a previous study showing E2 increases HSC proliferation and frequency and erythropoiesis (Nakada et al., 2014), showed E2 increases broad myelopoiesis, and uncovered the mechanistic link between E2 and UPR, a protective mechanism for HSCs (van Galen et al., 2014). They showed that IRE1-XBP1 branch of the UPR is responsible for E2's effect on HSCs, as deleting IRE1 using CRISPR in LSK cells blunted E2's effect on HSCs.1) Non-autonomous UPR refers to the phenomenon that ER stress in one tissue induces UPR in a remote tissue. If the authors claim that what they found here represents an example of non-autonomous UPR, they would need to show that ER stress in ovaries leads to release of E2, which activates UPR in HSCs. However, it does not seem to be the case here.

We have revised the text (Introduction, fourth paragraph and Discussion, last paragraph) to illustrate our viewpoint, which is that E2 is a systemic factor that activates UPR in HSPCs, and removed any claims that implied non-autonomous UPR in HSCs.

2) The observation that E2 increases myelopoiesis but not lymphopoiesis is interesting. Do myeloid primed HSCs have increased ER stress than lymphoid primed HSCs? The authors propose IRE1 as an underlying mechanism. However, in Figure 5, it appears that the effects of E2 and IRE1α are uncoupled. E2 increased WBC but IRE1α deletion had no effect on WBC. E2 had no effect on RBC but IRE1α deletion reduced RBC.

Using CD150 as a proxy for myeloid- and lymphoid-biased HSCs, we fractionated CD48^-^ LSK cells and examined whether Ire1α is induced in a particular subpopulation of HSCs (for example, myeloid-biased CD150-high HSCs). We observed similar increases of Ire1α mRNA levels in HSCs regardless of the CD150 expression levels, suggesting that E2 is not acting upon a subset of the HSC pool. This new data is provided in Figure 4—figure supplement 2. In regards to the second part of the comment, we included additional replicates to the data illustrated in Figure 6 (previously Figure 5), which allowed for a more robust and convincing interpretation of the data. Importantly, our new results shown in Figure 6 demonstrates that E2 promotes the recovery of immature HSPCs in the bone marrow of recipient mice, and this recovery depends clearly upon Ire1α. We believe that assessing HSPCs is a more direct way to examine hematopoietic regeneration, since the number of blood cells in peripheral blood can be regulated by multiple homeostatic mechanisms.

3) The authors performed RNA-seq and found that UPR genes are upregulated in E2-treated HSCs. Importantly, they showed that the UPR upregulation upon E2 is dependent on ERα. How does E2-ERα upregulate IRE1? Does IRE1 promoter have ERα binding site? Does ERα bind to IRE1 promoter?

For our full response to this comment, please see main concern point #2 above. As discussed above, we have provided a large amount of new data showing that ERα associates with the upstream region of Ire1α in E2 treated HSPCs. These data are provided in the new Figure 5.

4) In Figure 5, what's the editing efficiency? Typically, a pool of CRISPR edited cells do not show knockdown, but need to be selected for single clones that have efficient knockdown. Since the novel aspect of this study is the connection of E2 with UPR and the knockout mouse models for all three branches of UPR are well established, it makes more sense to use the KO mice.

For our full response, please see main concern point #1 above. A clonal analysis of Ire1α-editing efficiency was conducted on LSK cells. As seen in Figure 6—figure supplement 1 high proportion (~75%) of LSKs are edited using our previously established protocol, consistent with the immunoblotting data provided in Figure 6. We agree with the reviewer that selecting clones with efficient knockout is a robust strategy to study the effects of gene ablation. Multiple clones should be selected to avoid potential clone-specific phenotypes. This approach is technically difficult to perform with HSPCs, since these cells cannot be maintained for a long period of time, which is needed to select single clones by antibiotics resistance or fluorescent marker expression. Our approach takes advantage of the high efficiency of editing (~75%) in the bulk population, enabling us to examine the phenotypic changes in the bulk population, which has been taken by other groups as well (see response to main concern #1). It is true that there will be a small fraction of cells that were not gene ablated in our approach, which might explain the modest effects we observed in Figure 5 of the original manuscript. To address this issue, we have repeated the experiments and we observed statistically significant effects of Ire1α ablation in all three parameters we tested (WBC, RBC, PLT), as shown in the new Figure 6. As mentioned in response to comment #2, we have added new data showing that E2 promotes the recovery of immature HSPCs in the bone marrow of recipient mice, and this recovery depends clearly upon Ire1α.

5) The effect of E2 on hematopoiesis is inconsistent throughout the manuscript. e.g. in Figure 2 significantly increases platelets. However, in Figure 5 has no effect on platelets. It is unclear how robust the effect of E2 is.

As mentioned above, we have included additional experimental replicates to the figures in question. These additional replicates clarify the effects observed on hematopoietic regeneration. Furthermore, we have analyzed HSPCs in the bone marrow of the recipient mice (reviewer 2, comment 3). These additional data provided in Figure 6 are in agreement with our findings based on blood cell counts (Figure 6), and demonstrate that E2 enhances recovery of HSPCs in an Ire1α-dependent manner. We postulate that since production of mature cells (such as platelets) involves multiple intermediate cell stages during differentiation and that it is also affected by a variety of homeostatic mechanisms (such as thrombopoietin production by the liver and the kidney), the end output (CBC data) may not be as robust as directly examining HSPCs.

6) Figure 5—figure supplement 1 is missing.

We apologize for this oversight. We have included the missing figure supplements in the revised manuscript.

[Editors' note: further revisions were requested prior to acceptance, as described below.]

The manuscript has been improved but there are some remaining issues that need to be addressed before acceptance, as outlined below:1) Subsection “Estradiol promotes protective unfolded protein response in HSCs”, third paragraph. Please state clearly that clonal analysis of electroporated LSK cells indicated that about 25% of the clones were deleted (Figure 6—figure supplement 1). Nevertheless, bulk-modified cell populations used for subsequent experiments showed that….

We added this description that reads “Clonal analysis of electroporated LSK cells indicated that about 25% of the clones had deletions between the two sgRNA target sites, while another 50% of the clones had small indels detected by TIDE analysis (Figure 6—figure supplement 1). The bulk-modified cells were then…”

2) Subsection “Estradiol activates the unfolded protein response (UPR) pathway in HSCs”, first paragraph. In line with the detailed explanation presented in the response letter, please modify the text to indicate that a number of Myc target genes were induced by E2, even though c-Myc, N-Myc and L-Myc were not. It would be helpful to include the comments provided in this section of the response letter within the Discussion section of the manuscript.

We added the following sentence: “Several Myc target genes were induced by E2 treatment in vitro (Figure 4—figure supplement 1), although c-Myc, N-Myc, and L-Myc were not induced in HSCs by E2 as determined by RNA-seq (data not shown).”

Additionally, we added a new paragraph discussing the changes in Myc targets without induction of Myc themselves (Discussion, third paragraph).

3) Please indicate at least in the Materials and methods that, based on Figure 1—figure supplement 1, HSCs were sorted as CD48-/low (and for brevity named CD48-) according to Kiel et al., Oguro et al. etc. There is much confusion in the field about where to place CD48 gates, and the more clarity is used in the presentations, the better. Overall, the gating schemes should include the experimental treatments and not just the control as presented in Figure 1—figure supplement 1. This is particularly true for readers who are evaluating how the E2 and IR treatment conditions change distribution of markers such as CD48 and CD150.

We modified the methods (subsection “Flow-cytometry and HSC isolation”) to indicate that HSCs are defined as CD150^+^CD48^−/low^Lineage^−^Sca-1^+^c-kit^+^ cells, citing Kiel et al. and Oguro et al. We also replaced “CD48- “to “CD48-/low” throughout the manuscript, including Figure 4—figure supplement 2. We also added the gating schemes of the E2 treated mice in Figure 1—figure supplement 1. We also added gating schemes of LSK cells with or without irradiation, in Figure 3—figure supplement 1 (we did not use CD150 or CD48 in irradiation studies, as described already in subsection “Estradiol accelerates hematopoietic progenitor cell regeneration after ionizing radiation”).

4) Please include the comments about variable effects on platelets production presented in the response letter within the Discussion section.

We added this new discussion to the Discussion section.